# Macrophage-dependent IL-1β production induces cardiac arrhythmias in diabetic mice

Gustavo Monnerat[1,*], Micaela L. Alarcón[1,*], Luiz R. Vasconcellos[2,3], Camila Hochman-Mendez[1], Guilherme Brasil[1], Rosana A. Bassani[4], Oscar Casis[5], Daniela Malan[6], Leonardo H. Travassos[2], Marisa Sepúlveda[7], Juan Ignacio Burgos[7], Martin Vila-Petroff[7], Fabiano F. Dutra[3], Marcelo T. Bozza[3], Claudia N. Paiva[3], Adriana Bastos Carvalho[1], Adriana Bonomo[3,8], Bernd K. Fleischmann[6], Antonio Carlos Campos de Carvalho[1,9] & Emiliano Medei[1,9]

Diabetes mellitus (DM) encompasses a multitude of secondary disorders, including heart disease. One of the most frequent and potentially life threatening disorders of DM-induced heart disease is ventricular tachycardia (VT). Here we show that toll-like receptor 2 (TLR2) and NLRP3 inflammasome activation in cardiac macrophages mediate the production of IL-1β in DM mice. IL-1β causes prolongation of the action potential duration, induces a decrease in potassium current and an increase in calcium sparks in cardiomyocytes, which are changes that underlie arrhythmia propensity. IL-1β-induced spontaneous contractile events are associated with CaMKII oxidation and phosphorylation. We further show that DM-induced arrhythmias can be successfully treated by inhibiting the IL-1β axis with either IL-1 receptor antagonist or by inhibiting the NLRP3 inflammasome. Our results establish IL-1β as an inflammatory connection between metabolic dysfunction and arrhythmias in DM.

[1] Institute of Biophysics Carlos Chagas Filho, Universidade Federal do Rio de Janeiro, Rio de Janeiro 21941-902, Brazil. [2] LIRS-Laboratory of Immunoreceptors and Signaling, Universidade Federal do Rio de Janeiro, Rio de Janeiro 21941-902, Brazil. [3] Instituto de Microbiologia, Universidade Federal do Rio de Janeiro, Rio de Janeiro 21941-902, Brazil. [4] Center for Biomedical Engineering, University of Campinas, Campinas 13.083-970, Brazil. [5] Departamento de Fisiología, Facultad de Farmacia, Universidad del País Vasco UPV/EHU, 01006 Vitoria, Spain. [6] Institute of Physiology I, Life and Brain Center, University of Bonn, Bonn D-53127, Germany. [7] Centro de Investigaciones Cardiovasculares, Conicet La Plata, Facultad de Ciencias Médicas, Universidad Nacional de La Plata, La Plata 1900, Argentina. [8] FIOCANCER/ VPPLR/FIOCRUZ, FIOCRUZ-Manguinhos, Rio de Janeiro 21040-360, Brazil. [9] National Center for Structural Biology and Bioimaging—CENABIO/UFRJ, Rio de Janeiro 21941-902, Brazil. * These authors contributed equally to this work. Correspondence and requests for materials should be addressed to E.M. (email:emedei70@biof.ufrj.br).

Sterile inflammation usually resolves an initial insult[1], but sometimes persists, as in the case of metabolic dysfunction. This inflammation has been implicated in the pathogenesis of several metabolism-associated diseases and represents a putative link between diabetes mellitus (DM)[1–4] and secondary heart disease.

In diabetic patients, cardiomyopathy and its main complication, ventricular arrhythmias, are the leading cause of death[5,6]. In fact, diabetic patients show higher frequency of fatal ventricular arrhythmias and sudden cardiac death[7–10]. Moreover, these patients present both QT and corrected QT interval (QTc) prolongation due to increased ventricular action potential duration (APD), which predisposes to arrhythmogenesis[7,8,11]. The main mechanisms underlying these diabetic arrhythmias are still unknown, but we hypothesized that sterile inflammation triggered by hyperglycemia is the main pathophysiological mechanism. In fact, hyperglycemia upregulates toll-like receptor 2 (TLR2) expression in monocytes[12] leading to continuous interleukin (IL)-1β production, which in turn has been implicated as a mediator of the deleterious effects of hyperglycemia[13]. Additionally, the presence of this cytokine has been reported in type 1 diabetic hearts[14].

The nucleotide-binding domain and leucine-rich repeat containing protein (NLRP) family of cytosolic pattern recognition receptors has a critical role in promoting sterile inflammation. Recognition of danger signals released from failing or malfunctioning cells, leading to production of mature IL-1β, is a key event in this process[15]. TLR agonists can transcriptionally induce the expression of NLRP3 and pro-IL-1β[16]. NLRP3 oligomerizes with the adaptor molecule 'apoptosis-associated speck-like protein containing a CARD domain' (ASC) in response to different signals[1,17,18]. NLRP3 and ASC subsequently recruit the cysteine protease pro-caspase-1 to generate a caspase-1-activating platform, known as the inflammasome[1,19]. The inflammasome then promotes caspase-1-dependent proteolytic cleavage of pro-IL-1β into mature IL-1β. However, the effects of IL-1β on target organs, which are involved in the pathogenesis of DM, are still to be unravelled.

Here we investigated the role of sterile inflammation in the induction of arrhythmias in DM. We show that IL-1β produced by DM heart macrophages targets cardiomyocytes to induce cardiac arrhythmias. Using a multi methodological approach, we demonstrate how TLR2 and NLRP3 inflammasome activation in macrophages mediate the production of IL-1β. Our results indicate that the inflammatory response to the metabolic dysfunction in DM generates cardiac arrhythmias. Our study also reveals a novel treatment option of DM-related arrhythmias by using either an IL-1β receptor antagonist (anakinra) or a NLRP3 inhibitor (MCC-950).

## Results

**TLR2 is required for diabetes-induced arrhythmias.** Hyperglycemia upregulates TLR2 expression in monocytes[12] and TLR2 participates is cardiac sterile inflammation in a number of situations[20,21]. To investigate the involvement of the TLR2-IL-1β axis in cardiac electrical activity, DM was induced in wild-type (WT) and $Tlr2^{-/-}$ mice (Fig. 1a). Despite similar high blood glucose levels (inset Fig. 1a), the diabetic $Tlr2^{-/-}$ group presented shorter QT (Fig. 1b and Supplementary Fig. 1a,b) and QTc intervals, when compared with diabetic WT mice (Fig. 1c). In diabetic WT mice cardiac action potential (AP) displayed slower repolarization, when compared with non-diabetic WT mice, while in diabetic $Tlr2^{-/-}$ mice AP duration (APD) was similar to non-diabetic mice (Fig. 1d,e). The absence of TLR2 drastically reduced the susceptibility to cardiac arrhythmias after caffeine and dobutamine (Caff/Dobu) challenge in diabetic mice (Fig. 1f,g), which were not detected in non-diabetic animals (Supplementary Fig. 1d). These results were gender-independent, since these key findings were very similar in females (Supplementary Fig. 2). Cardiac relative mass and left ventricular morphology and function were preserved in all groups (Supplementary Fig 3a–e). No differences were observed in QRS duration or in cardiac fibrosis among experimental groups (Supplementary Fig. 1c and 3f,g). In a similar manner compared with an earlier study[22], we have observed a trend towards lower insulin levels in $Tlr2^{-/-}$ diabetic compared with $Tlr2^{-/-}$ non-diabetic mice (unpaired t-test; $P = 0.07$) (Supplementary Table 1), although the difference did not reach statistical significance. In addition, the $Tlr2^{-/-}$ diabetic mice showed insulin levels similar to WT + DM mice suggesting that: (i) the anti-inflammatory protection afforded by the lack of TLR2 was not enough to protect the streptozotocin-impaired pancreatic function; and (ii) the lack of electrophysiological changes observed in this KO mice could be not related to changes in insulin levels. TLR2 activation has a key role in metabolic and cardiac diseases and its presence is required to trigger IL-1β production in different animal models[23]. We observed that diabetic $Tlr2^{-/-}$ mice have lower systemic (Fig. 1h) and local (heart) (Fig. 1i,j and Supplementary Fig. 4) concentrations of IL-1β than diabetic WT mice. No differences were observed in the cardiac IL-1β messenger RNA expression between WT and $Tlr2^{-/-}$ (Supplementary Fig. 5). Collectively, these data demonstrate that TLR2 contributes to IL-1β production, electrical disturbances and arrhythmias in a mouse model of DM.

**IL-1β induces cardiac electrical vulnerability.** Since the lack of TLR2 expression not only prevents AP prolongation and electrical vulnerability but also the DM-induced rise in local and systemic levels of IL-1β, we hypothesized that IL-1β is involved in these cardiac changes. We, therefore, investigated whether 24 h exposure to IL-1β is able to induce cardiac functional changes by measuring APD, ion currents and $Ca^{2+}$ handling in isolated rat cardiomyocytes. First, we found that IL-1β prolongs the APD (Fig. 2a–c), then we investigated its effect on a key repolarizing potassium current, namely the transient outward potassium current ($I_{to}$). Incubation of isolated ventricular cardiomyocytes in the presence of IL-1β, using pathophysiological levels (60 pg ml$^{-1}$, for more details please see Fig. 1h), for 24 h resulted in a ∼35% reduction of this current (Fig. 2d–f).

Several types of arrhythmia have been attributed, at least in part, to enhanced diastolic $Ca^{2+}$ leak from the sarcoplasmic reticulum (SR)[24]. Using confocal imaging, we examined the impact of IL-1β on SR $Ca^{2+}$ leak by monitoring $Ca^{2+}$ spark properties, which were increased after IL-1β exposure (Fig. 2g,h,j and Supplementary Fig. 6b–f). Accordingly, a modest but significant negative post-rest variation in the SR $Ca^{2+}$ content was observed only in IL-1β-treated myocytes (Fig. 2i). However, the SR $Ca^{2+}$ content was not significantly different in cells incubated for 24 h in the presence and absence of IL-1β (Supplementary Fig. 6g). Rest does not typically affect the SR $Ca^{2+}$ load, unless $Ca^{2+}$ efflux by the $Na^+/Ca^{2+}$ exchanger (NCX) is enhanced[25,26]. In this case, however, one would expect a greater ratio of the rate constants of $[Ca^{2+}]i$ re-uptake upon application of 10 mM caffeine or by electrical stimulation[27], in which $Ca^{2+}$ clearance from the cytosol is dependent mainly on the NCX and the SR $Ca^{2+}$ ATPase plus NCX, respectively[28]. However, neither the rate constants of $Ca^{2+}$ re-uptake/efflux (not shown), nor their ratio (Supplementary Fig. 6h) was significantly changed by IL-1β incubation, which indicates that the IL-1β-induced diastolic leak is not accompanied by stimulation of NCX-mediated $Ca^{2+}$ efflux.

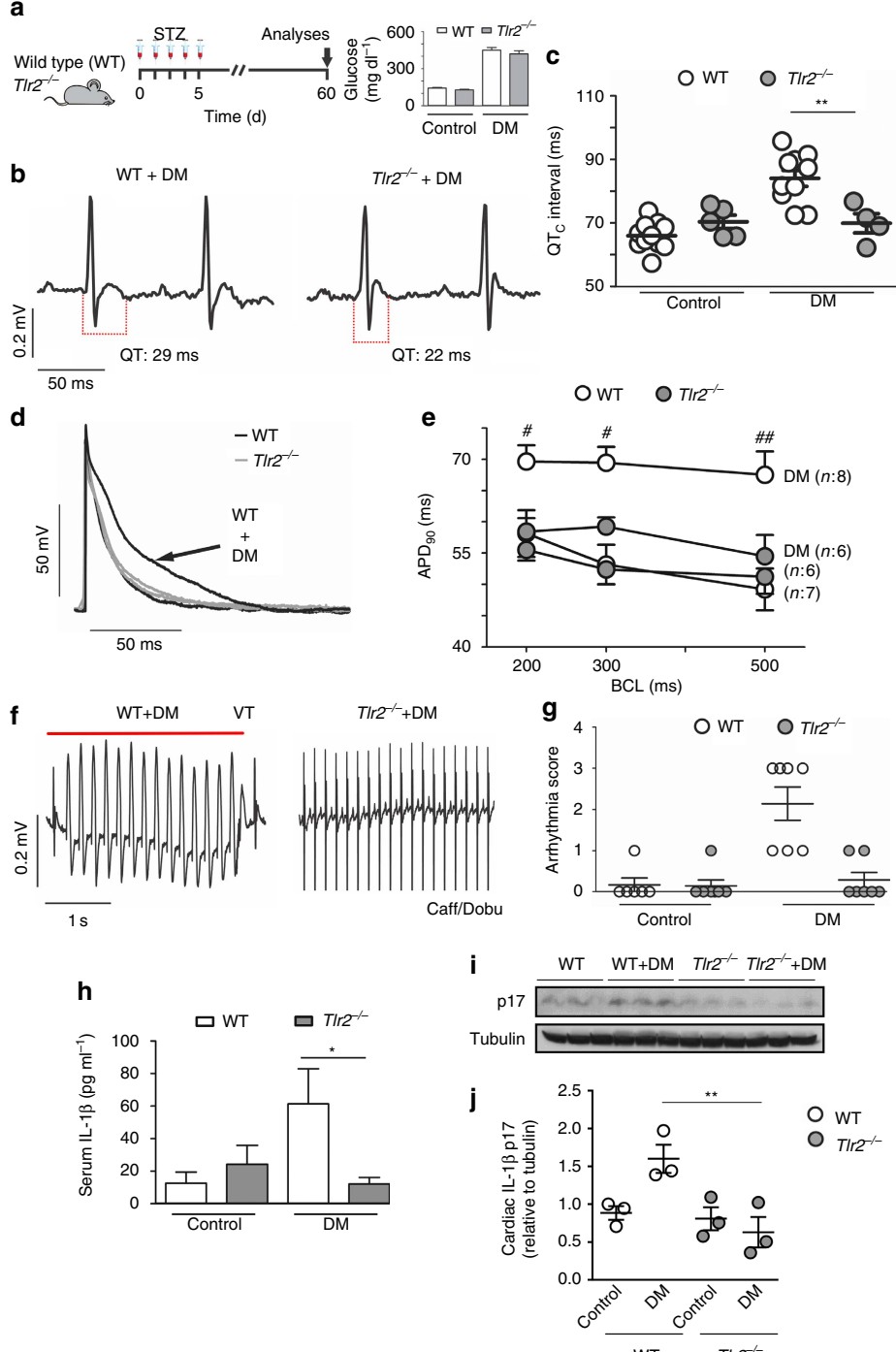

**Figure 1 | TLR2 regulates cardiac electrical parameters and incidence/severity of DM-induced arrhythmias. (a)** Experimental protocol: diabetes (DM) was induced in wild-type mice (WT) and toll-like receptor 2 *knock-out* mice ($Tlr2^{-/-}$) by five daily i.p. injections of streptozotocin (STZ) (50 mg kg$^{-1}$) and several parameters were analysed 60 days after the beginning of the protocol. Inset shows serum glucose levels of all experimental groups at day 60 ($n$ = WT: 10; $Tlr2^{-/-}$: 10; WT + DM: 10; $Tlr2^{-/-}$ + DM: 10). **(b)** Representative ECG traces of DM mice highlighting QT interval prolongation only in the WT group. **(c)** Corrected QT (QTc) interval duration ($n$ = WT: 10; $Tlr2^{-/-}$: 5; WT + DM: 10; $Tlr2^{-/-}$ + DM: 4). **(d)** Representative action potential (AP) traces from the endocardial layer of left ventricle strips at 300 ms basic cycle length (BCL) stimulation. **(e)** AP duration at 90 per cent of repolarization (APD$_{90}$) in different BCL ($n$ = WT: 7; $Tlr2^{-/-}$: 6; WT + DM: 8; $Tlr2^{-/-}$ + DM: 6). **(f)** Representative ECG traces during arrhythmia vulnerability test induced by caffeine and dobutamine (Caff/Dobu) showing ventricular tachycardia (VT - see red line) in WT + DM mice and a normal ECG in $Tlr2^{-/-}$ + DM. **(g)** Score quantification of arrhythmia incidence and severity ($n$ = WT: 6; $Tlr2^{-/-}$: 6; WT + DM: 7; $Tlr2^{-/-}$ + DM: 7). **(h-j)** Serum and local (heart) protein levels of IL-1β after 60 days of DM induction in experimental groups. Graph represents three (serum) and two (hearts, $n$ = 6 mice per group) independent experiments performed in duplicate. The results are expressed as mean ± s.e.m. Scatter plot shows values from individual mice, where horizontal bars represent means and error bars, s.e.m. * and ** represent respectively $P < 0.05$ and $P < 0.01$ versus WT + DM, (unpaired $t$-test). # and ## represent, respectively, $P < 0.05$ and $P < 0.01$ versus WT + DM (Bonferroni's *post* test following two-way ANOVA).

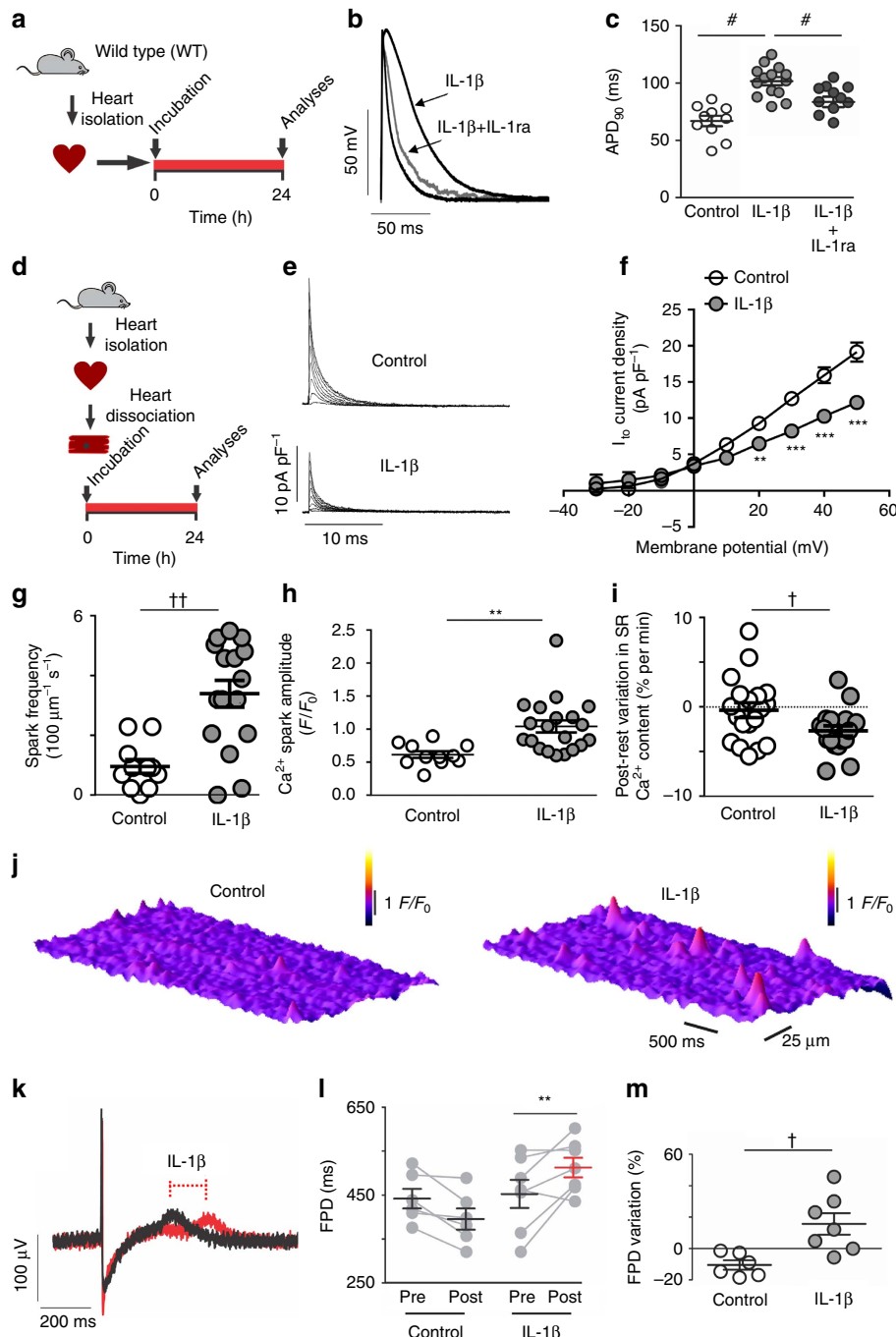

**Figure 2 | IL-1β induces electrical abnormalities in rat and human cardiac cells. (a)** Experimental protocol. Cardiac muscle strips from rat hearts were incubated for 24 h with IL-1β (10 ng ml$^{-1}$) with or without IL-1 receptor antagonist (IL-1ra: 100 ng ml$^{-1}$) and electrical function was assessed. (**b,c**) Representative action potential (AP) traces from left ventricle endocardial at 300 ms basic cycle length (BCL) before and after incubation with IL-1β (left panel), and APD$_{90}$ data from endocardial layer of left ventricular strips (right panel) (n of cells = Control: 10; IL-1β: 14; IL-1β + IL-1ra: 12; 4 hearts). (**d**) Experimental protocol: Hearts from rats were enzymatically dissociated and isolated cardiomyocytes were incubated for 24 h with IL-1β. (**e,f**) Representative traces (left panel) and current versus voltage (IV) curves (right panel) of I$_{to}$ current after exposure to IL-1β (n of cells = Control: 17; IL-1β (60 pg ml$^{-1}$): 5; from three hearts). (**g**) Quantification of calcium spark frequency and (**h**) spark fluorescence amplitude (F/F$_0$) (n of cells = Control: 11; IL-1β 20; cells from three hearts). (**i**) Post-rest decrease in sarcoplasmatic reticulum (SR) calcium content expressed as variation as per cent of the steady-state content per min rest (n of cells = Control: 19; IL-1β 18; four hearts). (**j**) Representative images of calcium spark before and after incubation or not with IL-1β. (**k**) Representative traces of field potential in human iPS-derived cardiomyocytes highlighting effect of IL-1β on the field potential duration (FPD. (**l,m**) FPD and its variation in individual cell preparations before and after 24 h of incubation with control solution or IL-1β (10 ng ml$^{-1}$) (n = Control: 6; IL-1β: 7; from three independent experiments). The results are expressed as mean ± s.e.m. Scatter plot shows values from individual cell preparations, where horizontal bars represent means and error bars, s.e.m. $^\#$ represents $P < 0.05$ (Bonferroni's *post* test following one-way ANOVA). ** and *** represent, respectively, $P < 0.01$ and $P < 0.001$ (Bonferroni's *post* test following two-way ANOVA). † and †† represent, respectively, $P < 0.05$ and $P < 0.01$ (unpaired *t*-test).

We next investigated whether the reported IL-1β effects on cardiac electrical properties in rodents could be also observed in human cardiomyocytes. We, therefore, tested IL-1β effects in human induced pluripotent stem-cell derived cardiomyocytes (hIPS-CM) using field potential duration (FPD) measurement. Our experiments revealed longer FPD on IL-1β treated hIPS-CM, when compared with untreated hIPS-CM (Fig. 2k–m).

Thus, IL-1β decreases $I_{to}$ current, prolongs repolarization, and increases diastolic SR $Ca^{2+}$ leak, providing a cellular substrate for triggered arrhythmogenic activity.

**IL-1β and pCaMKII/oxiCaMKII induce arrhythmias.** $Ca^{2+}$ and calmodulin-dependent protein kinase II (CaMKII) have been implicated in diabetes-associated arrhythmias[29,30] and recent evidence indicates that MyD88 inflammatory signalling leads to CaMKII oxidation and activation[31]. We, therefore, explored CaMKII activity upon exposure of cardiac strips to 10 ng ml$^{-1}$ of IL-1β for 24 h and then freeze-clamped the tissue for western blotting. IL-1β effectively promoted CaMKII oxidation (oxiCaMKII) and phosphorylation (p-CaMKII) (Fig. 3a,b and Supplementary Fig. 7). To determine whether mere exposure to IL-1β is sufficient to generate a pro-arrhythmogenic substrate, ventricular myocytes were incubated for 24 h with IL-1β and the number of spontaneous contractile events (NSE), an index of cellular arrhythmic activity, was measured after pacing the cells during 10 min at either 0.5 or 3 Hz. Cells incubated in the absence of IL-1β (control) had only sporadic NSE, whereas they were significantly enhanced by IL-1β (Fig. 3c,d). Importantly, the IL-1β-induced NSE were largely reduced when tissue was pretreated with 2.5 µmol l$^{-1}$ of the CaMKII inhibitor, KN93 (Fig. 3c). Since it is well accepted that KN-93 has numerous off target actions, including ionic currents with potential relevance to the observed arrhythmia phenotypes, a set of experiments were conducted using, the KN93 inactive analog KN92, either in the absence or presence of IL-1β (Fig. 3d). While KN92 alone did not significantly affect the NSE, the combination of KN92 + IL-1β induced an increase in the number of NSE. In order to confirm the role of CaMKII as a mediator of the increased NSE, another set of experiments were conducted using cardiomyocytes from mice with genetic CaMKII inhibition by cardiac-specific expression of autocamtide 3 inhibitory peptide (AC3-I) mimicking a conserved sequence of the CaMKII regulatory domain and their respective controls, which have an scrambled version of AC3-I (autocamtide 3 control peptide (AC3-C))[32]. Such as shown in Fig. 3c,d, AC3-C cells exposed to IL-1β presented a higher number of NSE compared with the same cell type under control condition. Conversely, the AC3-I cells exposed to IL-1β did not show a significant increase in NSE (Fig. 3c,d and Supplementary Fig. 6). Additionally, in order to elucidate whether the CaMKII inhibition also could have a key role in human cells, hIPS-CM were exposed to IL-1β in the absence or in the presence of a selective CaMKII inhibitor (autocamtide—2—related inhibitory peptide (AIP)). In this setting, it was observed that the CaMKII inhibition was able to prevent the IL-1β-induced longer FPD in hIPS-CM (Fig. 3e,f).

Collectively these data demonstrated the key role-played by CaMKII activation in the electrophysiological changes induced by IL-1β.

**IL-1β production by TLR2-activated macrophages in heart tissue.** In order to demonstrate that TLR2 stimulation was involved in IL-1β release, we incubated left ventricular myocardial strips from healthy rats for 24 h with a TLR2/1 agonist (Pam3) (Fig. 4a); these strips are known to contain other cell types besides CMs, such as immune cells. Pam3 incubation

yielded a significant increase in cardiac IL-1β concentration (Fig. 4b). Since IL-1β secretion can be triggered by TLR2 activation in macrophages[23], a group of animals was previously treated with liposomes containing clodronate-L (dichloromethylene diphosphonate ($Cl_2MDP$)), a chemical agent that induces macrophage apoptosis[33], in order to test whether macrophages are the source of IL-1β production induced by TLR2 activation. As a control condition, liposomes containing PBS were used (Fig. 4a,b). Stimulation of myocardial strips from macrophage-depleted animals with Pam3 resulted in significantly lower IL-1β amounts compared with those from non-macrophage-depleted healthy hearts (PBS liposome condition) (Fig. 4b). In addition, stimulation of the left ventricular wall for 24 h with either Pam3 or IL-1β promoted similar cardiac AP prolongation, an effect prevented by simultaneous incubation with IL-1 receptor antagonist (Figs 2a–c and 4c–e). Macrophage depletion prevented not only the rise in IL-1β (Fig. 4b), but also the AP prolongation seen after Pam3 stimulation (Fig. 4f,g). Conversely, the hearts obtained from animals pre-treated with liposome-PBS (control condition) showed longer AP repolarization after 24 h incubation with Pam3 (Fig. 4f,g). Taken together, these results demonstrate that the APD prolongation induced by TLR2 stimulation is mediated by cardiac macrophages through IL-1β production.

**$Tlr2^{-/-}$ have lower MHCII$^{high}$ and NLRP3 macrophage content.** Recent publications described the importance of MHCII$^{high}$ cardiac macrophages for IL-1β secretion in different cardiac diseases[34]. We assessed the presence of macrophage subsets in hearts of diabetic mice (Fig. 5a–c and Supplementary Fig. 8). The percentage of MHCII$^{high}$ macrophages was reduced in diabetic $Tlr2^{-/-}$ mice compared with diabetic WT mice, despite a similar percentage in the hearts of non-diabetic $Tlr2^{-/-}$ and WT animals, indicating that, in the absence of TLR2, diabetic mice display a reduction in the cardiac macrophage subpopulation responsible for the IL-1β production. TLR2 expression in cardiac macrophages was similar in non-diabetic and diabetic WT mice (Supplementary Fig. 9).

Several groups have described the importance of NLRP3 inflammasome activation in macrophages for IL-1β maturation/production in DM[3,35]. Therefore, we also investigated NLRP3 content (red) in resident macrophages (F4/80 marker, green) present in cardiac tissue from WT and $Tlr2^{-/-}$ control and diabetic mice. We observed lower NLRP3 content in macrophages from diabetic $Tlr2^{-/-}$ mice compared with macrophages from diabetic WT mice (Fig. 5d,e), indicating that NLRP3 expression is upregulated by TLR2 signalling.

**Activation of NLRP3 inflammasome and arrhythmias in DM.** As we have shown that cardiac macrophages produce IL-1β and upregulate NLRP3 expression in response to TLR2 stimulation promoted by DM, we tested whether the NLRP3 inflammasome is involved in the DM-induced cardiac electrical disturbances and susceptibility to arrhythmias. For this purpose, DM was induced in $Nlrp3^{-/-}$ and $Casp1^{-/-}$ as well as in WT mice (Fig. 6a,b). $Nlrp3^{-/-}$ diabetic mice showed insulin levels similar to $Nlrp3^{-/-}$ non-diabetic mice (unpaired $t$-test; $P = 0.17$), despite their increased glycemia levels. However, $Casp1^{-/-}$ diabetic mice presented a trend towards lower levels of insulin, than $Casp1^{-/-}$ non-diabetic mice ($P = 0.057$; Supplementary Table 2). These results suggest that while the electrophysiological improvement observed could not be related to the rescue of the insulin levels in the $Casp1^{-/-}$ diabetic mice, we cannot exclude that insulin could be also involved in the electrophysiological improvement observed in $Nlrp3^{-/-}$ diabetic mice. Both heart

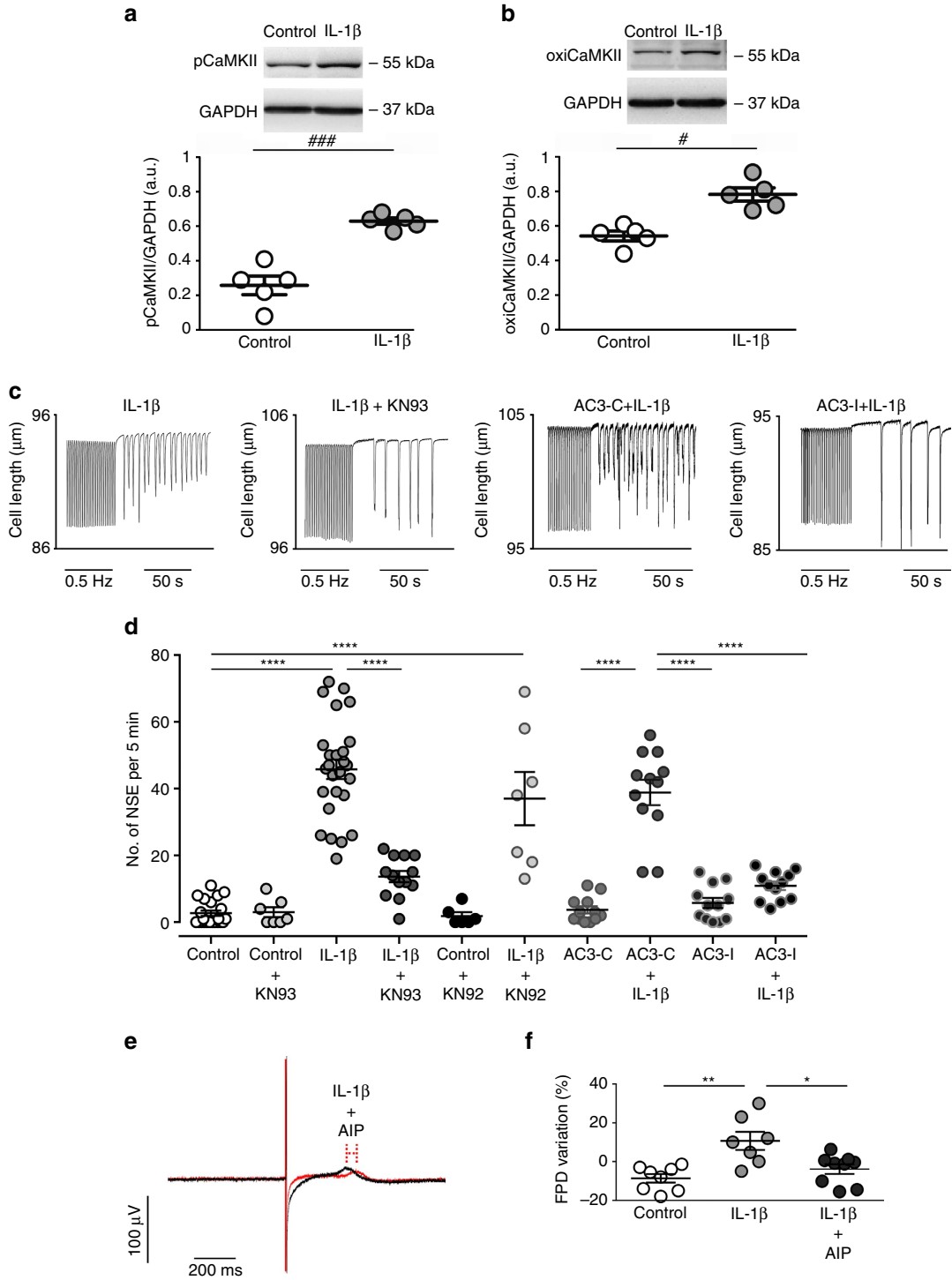

**Figure 3 | IL-1β induces pCaMKII/oxiCaMKII and spontaneous cardiac activity. (a)** Representative immunoblots of pCaMKII and GAPDH and quantitative densitometry values after 24 h incubation in the absence (control) or presence of IL-1β. **(b)** Representative immunoblots of oxiCaMKII and GAPDH and quantitative densitometry values. **(c)** Representative traces of cell shortening in isolated rat cardiomyocyte after 24 h incubation with IL-1β or IL-1β plus CaMKII inhibitor (KN93) as well as in transgenic mice with myocardial-delimited expression of the specific peptide inhibitor of CaMKII inactive control (AC3-C) or the active inhibitory (AC3-I), in which spontaneous contractions developed after pacing (0.5 Hz) interruption. **(d)** Number of spontaneous contractions for 5 min after pacing (NSE), which was increased after IL-1β incubation ($n$ of cells = Control: 22; Control + KN93: 7; IL-1β: 26; IL-1β + KN93: 13; Control + KN92: 6; IL-1β + KN92: 7; AC3-C: 12; AC3-C + IL-1β: 12; AC3-I: 12; AC3-I + IL-1β: 12; from three hearts). **(e)** Representative traces of field potential in human iPS-derived cardiomyocytes highlighting inhibitory influence of CamKII with AIP of the IL-1β effect on the field potential duration (FPD). **(f)** FPD variation in individual cell preparations after 24 h incubation with control solution or IL-1β + AIP (10 ng ml$^{-1}$ and 1 µmol l$^{-1}$, respectively) ($n$ = Control: 8; IL-1β: 7; IL-1β + AIP: 9; obtained from three independent experiments). The results are expressed as mean ± s.e.m. Scatter plot shows values from individual cell preparations, where horizontal bars represent means and error bars, s.e.m. # and ### represents, respectively, $P < 0.05$ and $P < 0.001$ (unpaired $t$-test). *, ** and **** represent, respectively $P < 0.05$, $P < 0.01$ and $P < 0.0001$ (Bonferroni's $post$ test following one-way ANOVA).

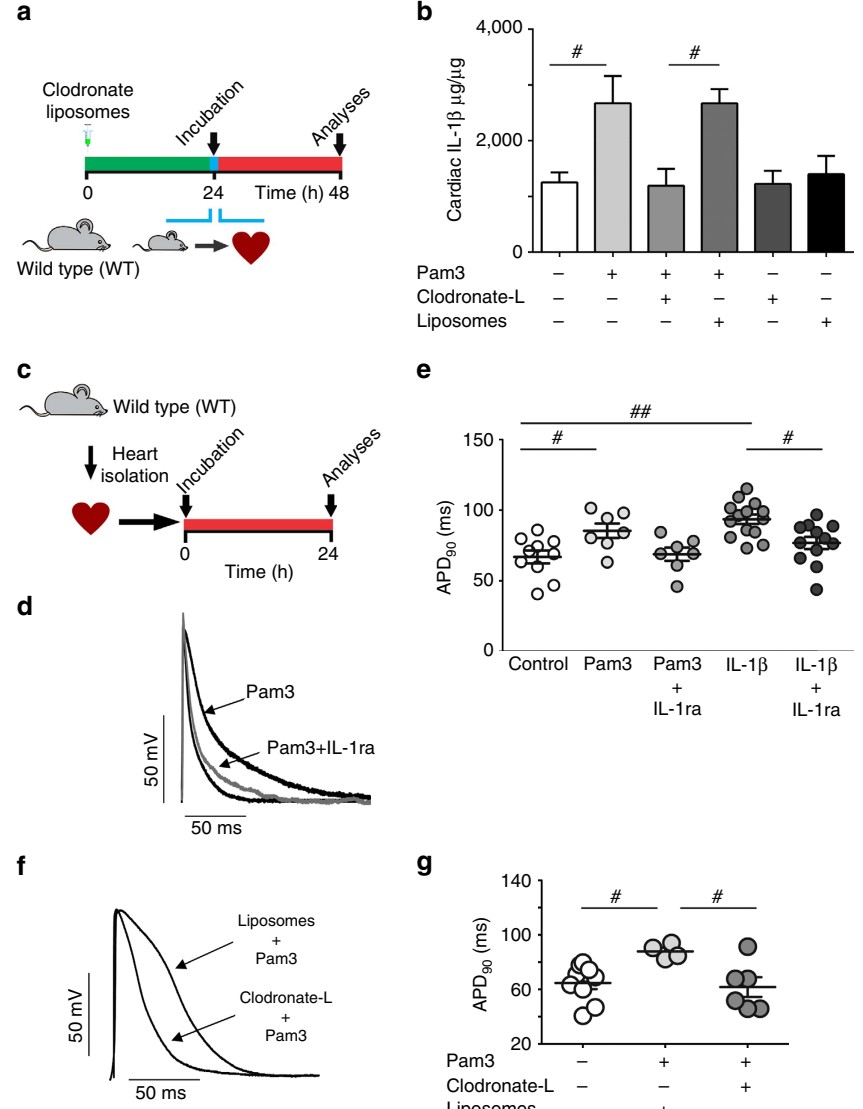

**Figure 4 | TLR2 regulates APD through production of IL-1β by cardiac macrophages. (a)** Experimental protocol: macrophages from rat hearts were depleted with clodronate liposomes (clodronate-L). After depletion, cardiac muscle strips were incubated with TLR2 agonist Pam3 (1 µg ml$^{-1}$). Liposomes containing PBS were used as control vehicle solution (Liposomes). **(b)** Enzyme-linked immunosorbent assay quantification of IL-1β cardiac content (graph represents three independent experiments in duplicate, $n = 6$ hearts per group). **(c)** Experimental protocol: rat cardiac muscle strips were incubated for 24 h with Pam3 (1 µg ml$^{-1}$) in the presence or absence of the IL-1 receptor antagonist IL-1ra (100 ng ml$^{-1}$) and electrical function was assessed ($n$ of cells = Control: 10; Pam3: 7; Pam3 + IL-1ra: 7; IL-1β: 14; IL-1β + IL-1ra: 12; from four hearts). **(d)** Representative action potential (AP) traces from endocardial layer of left ventricle strips at 300 ms basic cycle length (BCL). **(e)** The graph summarizes the APD$_{90}$ values from endocardial layer of left ventricle strips at 300 ms BCL under different condition. **(f,g)** Representative AP traces (left panel) and APD$_{90}$ values (right panel) in rats cardiac strips from animals pre-treated with clodronate liposomes or control liposomes and then exposed 24 h to Pam3 ($n$ = Control: 10; Pam3 + liposomes: four hearts ; Pam3 + clodronate-L: six hearts). The results are expressed as mean ± s.e.m. Scatter plots show values from individual cell preparations, where horizontal bars represent means and error bars, s.e.m. # and ## represents, respectively $P < 0.05$ and $P < 0.01$ (Bonferroni's *post* test following one-way ANOVA).

weight/body weight or heart weight/tibia length ratios were similar among study groups (Supplementary Table 2). The absence of NLRP3 or Casp1 prevented DM-induced QTc (Fig. 6c) and AP prolongation (Fig. 6d,e). Importantly, $Nlrp3^{-/-}$ and $Casp1^{-/-}$ diabetic mice had lower arrhythmia vulnerability and severity in response to Caff/Dobu challenge when compared with diabetic WT mice (Fig. 6f,g). The $Nlrp3^{-/-}$ and $Casp1^{-/-}$ non-diabetic mice showed cardiac electrical profile similar to WT non-diabetic mice (Supplementary Fig. 10a–e). Since the lack of NLRP3 prevented DM-induced cardiac changes, we tested next whether NLRP3 inhibition could reverse diabetes-induced cardiac electrical disturbances (Fig. 6h). The IL-1β release inhibitory compound CRID3 (CP-456773 (refs 36,37), renamed MCC-950) is found to inhibit the NLRP3 inflammasome by unknown mechanisms[35]. After 15 days of treatment, MCC-950 did not interfere with high glucose levels and reversed the prolongation of QT and QTc intervals in WT diabetic mice (Fig. 6i–k and Supplementary Fig. 10f,g). This was fully in accordance with the electrical vulnerability testing, as mice treated with MCC-950 had significantly lower arrhythmia vulnerability and severity compared with saline-treated WT diabetic control mice (Fig. 6l,m). Thus, activation of the NLRP3 inflammasome is required for diabetes to induce prolonged QTc and arrhythmias.

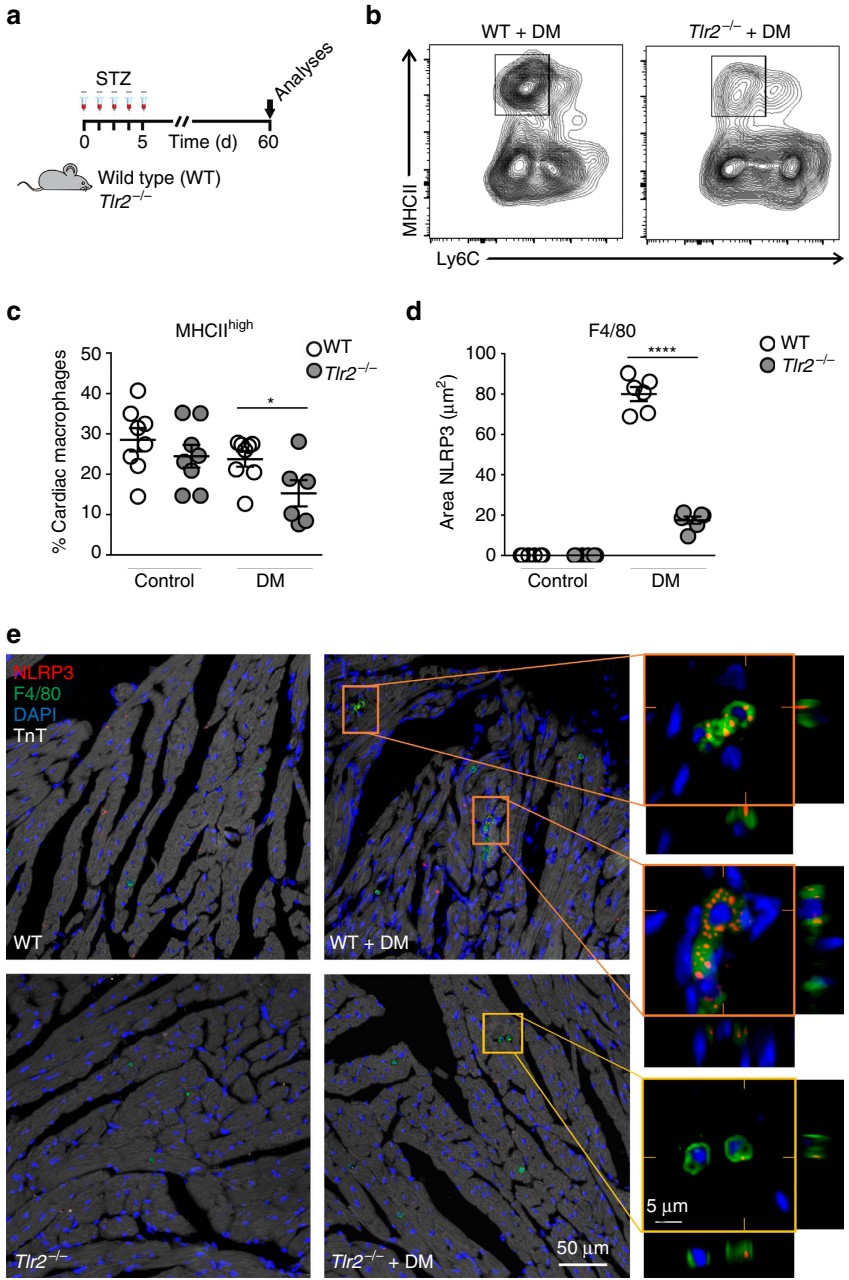

**Figure 5 | TLR2 regulates macrophage subsets and NLRP3 expression in diabetic hearts.** (**a**) Experimental protocol. (**b**) Flow cytometry of cardiac macrophages shows decrease of MHCII Ly6C double-positive macrophages in the $TLR2^{-/-}$ + DM mouse heart. (**c**) Percentage of cardiac macrophages positive for MHCII$^{high}$ and Ly6c (see Supplementary Fig. 8 for gating strategy) (n of hearts per group = WT: 7; $Tlr2^{-/-}$: 8; WT + DM: 8; $Tlr2^{-/-}$ + DM: 6). (**d**) Quantification of NLRP3 immunostained area in cardiac tissue shows increase of NLRP3 in cardiac macrophages from WT + DM mice (n of hearts per group = WT: 6; $Tlr2^{-/-}$: 6; WT + DM: 6; $Tlr2^{-/-}$ + DM: 6). (**e**) Representative immunostaining shows higher NLRP3 (red) content in cardiac (TnT—white) macrophages (F4/80—green) of WT + DM mice, but low expression in $TLR2^{-/-}$ + DM. The results are expressed as mean ± s.e.m. Scatter plots show values from individual mice, where horizontal bars represent means and error bars, s.e.m. * and **** represents, respectively $P < 0.05$ and $P < 0.0001$ (unpaired t-test).

**IL-1β has a key role in susceptibility to arrhythmias**. In order to unequivocally prove that IL-1β is responsible for DM-induced cardiac electrical changes, we induced DM in $IL-1r^{-/-}$ mice (Fig. 7a). $IL-1r^{-/-}$ developed hyperglycemia and lower insulin levels in response to streptozotocin (unpaired t-test; $P = 0.0001$ and $P = 0.04$ respectively; Supplementary Table 3; Fig. 7b), but in the absence of IL-1r no prolongation of QT (Supplementary Fig. 11a) and QTc intervals after DM induction (Fig. 7c) could be observed. Likewise, $IL-1r^{-/-}$ mice were less susceptible to

arrhythmia induction after the Caff/Dobu challenge (Fig. 7d,e). No differences in body weight/cardiac weight as well as the heart weight/tibia length ratios were observed between $IL-1r^{-/-}$ diabetic and non-diabetic mice (Supplementary Table 3).

These results strongly indicate that IL-1β signalling has a key role in the cardiac electrical disturbances observed in DM. We thus sought to explore potentially therapeutic effects of IL-1 receptor antagonist (IL-1Ra) to rescue the electrical cardiac function in diabetic WT mice. After 15 consecutive days of

IL-1Ra (Anakinra) administration ($10\,mg\,kg^{-1}$ per day intraperitoneal (i.p.)), even under sustained hyperglycemic state, diabetic IL-1Ra-treated mice presented much shorter QT and QTc intervals than non-treated diabetic mice (Fig. 7f–j—This experiment shares the same control group as Fig. 6h–k—and Supplementary Fig. 11). In addition, the IL-1Ra-treated diabetic mice showed lower arrhythmia vulnerability upon the Caff/Dobu challenge (Fig. 7k,l).

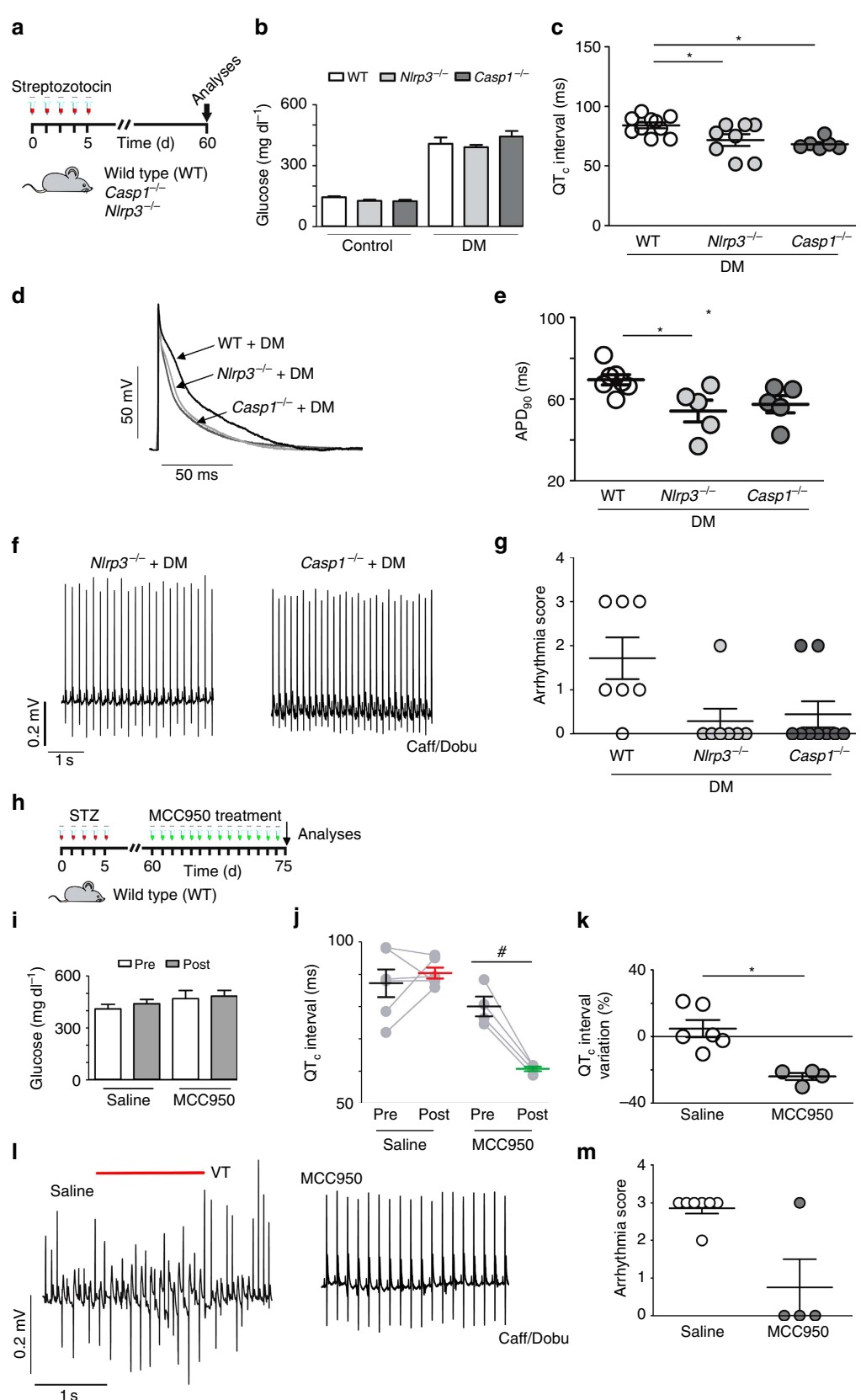

## Discussion

Diabetic patients often suffer of associated cardiovascular diseases, such as increased susceptibility to ventricular arrhythmias and sudden cardiac death. The mechanistic link between DM and arrhythmias, however, remains unknown. Here we dissected the complex signalling pathways involving different cell types present in the diabetic heart. DM induces a sterile inflammation that activates TLR2 and the NLRP3 inflammasome in heart macrophages to produce IL-1β. In cardiomyocytes, IL-1β then induces AP prolongation (through a decrease in $I_{to}$ current) and an increase in $Ca^{2+}$ sparks resulting in increased electrical vulnerability to arrhythmias. In the absence of TLR2, NLRP3, Caspase 1 or IL-1r, DM fails to sensitize rodents to ventricular arrhythmias. In addition, treatment with either MCC-950 (NLRP3 inhibitor) or anakinra (IL-1ra) reverses the cardiac electrical remodelling and decreases the incidence of arrhythmias, emphasizing the pathomechanistic relevance of IL-1β and also a novel pharmacological strategy for the treatment of DM-associated arrhythmias.

The activation of the inflammasome and the secretion of IL-1β seem to be a major danger-sensing pathway to sterile inflammation and cardiac repair in several heart diseases[38]. Earlier studies have demonstrated activation of the NLRP3 inflammasome and increased expression of IL-1β in both type 1 (ref. 39) and type 2 (refs 40,41) diabetes (T1D and T2D), but until now their relevance in the pathogenesis of diabetic heart disease has been unknown. Herein we show that genetic deletion of TLR2, NLRP3, Casp1 or IL-1r is sufficient to prevent both QTc prolongation and increased susceptibility to arrhythmias. Our results suggest that DM-triggered TLR2 activation induces the expression of pro-IL-1β and the NLRP3 activation in heart macrophages, thus promoting pro-IL-1β cleavage to a mature form by Casp1 resulting in IL-1β secretion. Then IL-1β acts on cardiomyocytes to prolong the AP and to increase $Ca^{2+}$ sparks, sensitizing hearts to arrhythmias. Although the endogenous TLR2 agonists and NLRP3 activators in our T1D model are not precisely known, hyperglycemia is the most likely suspect to trigger IL-1β production, since it is able to activate monocytes through TLR2 to produce IL-1β[12,42]. The intermediary steps from hyperglycemia to TLR2 and NLRP3 activation may involve the release of endogenous TLR2 ligands[43] and the generation of oxidative stress[44]. The increase in both TLR2 and IL-1β expression in diabetic patients[45–47] suggests that this mechanism of sterile inflammation is active in the natural course of disease.

A protective effect of TLR2 deficiency in DM has previously been observed by others[20], as well as the role of TLR2 activation in the induction of arrhythmic events in an ischemia/reperfusion animal model[21]. The close association between TLR2 activation and IL-1β secretion and their influence on arrhythmogenesis indicates that TLR2 and IL-1β are part of an inflammatory pathway that has a major role in the pathogenesis of diabetes-associated electric dysfunction of the heart.

The absence of insulin signalling can reduce $I_{to}$ current, resulting in AP and QT prolongation[48]. Nevertheless, the circulating insulin levels in diabetic $IL\text{-}1r^{-/-}$ were significantly decreased compared with non-diabetic counterparts, while a strong trend towards decreased levels was found in diabetic $Tlr2^{-/-}$ and $Casp1^{-/-}$ mice, even though it was not statically different. Only diabetic $Nlrp3^{-/-}$ mice presented circulation insulin levels similar to non-diabetic controls. Although we cannot exclude the possibility that preserved circulating insulin levels account for the prevention of electrophysiological abnormalities in diabetic $Nlrp3^{-/-}$ mice, this is certainly not the case in $IL\text{-}1r^{-/-}$, $Tlr2^{-/-}$ and $Casp1^{-/-}$ mice. These results reflect the limitation of our study where the systemic and local anti-inflammatory effects in the KO animals are not discernable using our global KO mice models.

The source of IL-1β production in vivo has been elusive in disease models in which inflammasome activation has a major pathophysiological role. In T1D patients, monocytes spontaneously secrete IL-1β[49], indicating that macrophages may represent such a source. Our results show that depletion of macrophages from cardiac muscle strips not only prevented TLR2 activation-induced IL-1β release, but also the Pam3/TLR2/IL-1β-induced prolongation of the cardiac AP, pointing to macrophages as the source of IL-1β secretion. We also show increased NLRP3 content inside cardiac macrophages of WT diabetic mice compared with $Tlr2^{-/-}$ diabetic mice, demonstrating that macrophage inflammasomes are activated in vivo.

The direct effects of IL-1β on electrical function of cardiomyocytes reported in the literature are scant. A recent review[50] on this subject highlights the rather indirect evidences linking IL-1, TNF and IL-6 to delayed repolarization, long QT syndrome and ventricular tachycardia. In 1993, IL-1 (not the mature IL-1β) was shown to increase the duration of the AP[51]. Another study described acute effects of IL-1β on atrial preparations, and did not find any effects on APD[52]. Herein we demonstrate that IL-1β creates a pro-arrhythmic environment both in rodent and human cells by reducing the repolarizing $K^+$ current ($I_{to}$), by increasing CaMKII oxidation/ phosphorylation and $Ca^{2+}$ spark frequency. A prolongation of the cardiac AP due to lower $I_{to}$ current has been described in diabetic experimental models[53,54] and in agreement with these data, we show here that binding of IL-1β to its receptor in the heart can produce similar effects. Hyperglycemia was previously reported to enhance CaMKII-dependent activation of spontaneous SR $Ca^{2+}$ release events[29]. We also provide evidence that IL-1β increases both $Ca^{2+}$ spark frequency and loss of SR $Ca^{2+}$ content during rest in cardiac cells, indicating enhanced SR $Ca^{2+}$ leak. Moreover, although the possibility that IL-1β could promote oxidation and activation of CaMKII has not been explored in cardiac tissue, recent evidence suggests that this could be achieved via MyD88 inflammatory signalling[31]. Since the IL-1β receptor uses MyD88 as a signalling adaptor and we illustrate that IL-1β promotes

**Figure 6 | DM-induced cardiac electrical alterations are regulated by the NLRP3 inflammasome.** (**a**) Experimental protocol: diabetes induction in WT, NLRP3 knockout ($Nlrp3^{-/-}$) and Casp1 knockout mice ($Casp1^{-/-}$). (**b**) Fasting blood glucose levels at day 60 (n = WT + DM: 10; $Nlrp3^{-/-}$ + DM: 8; $Casp1^{-/-}$ + DM: 10). (**c**) Corrected QT (QTc) interval duration (n = WT + DM: 10; $Nlrp3^{-/-}$ + DM: 8; $Casp1^{-/-}$ + DM: 6). (**d,e**) Representative action potential (AP) traces (left panel) and APD₉₀ data (right panel) from the endocardial layer of left ventricle strips at 300 ms basic cycle length (BCL) stimulation (n = WT + DM: 7; $Nlrp3^{-/-}$ + DM: 5; $Casp1^{-/-}$ + DM: 5). (**f**) Representative traces of arrhythmic vulnerability test induced by Caff/Dobu showing normal ECGs in $Nlrp3^{-/-}$ + DM and $Casp1^{-/-}$ + DM mice (n = WT + DM: 7; $Nlrp3^{-/-}$ + DM: 7; $Casp1^{-/-}$ + DM: 9). (**g**) Score quantification of arrhythmia incidence and severity in diabetic groups. (**h**) Experimental protocol for MCC-950 treatment (50 mg kg$^{-1}$ i.p./daily/15 consecutive days). (**i**) Fasting blood glucose levels pre (day 60) and post (day 75) treatment from at least four mice per group. (**j,k**) QTc interval values and per cent variation pre (60 days) and post (75 days) treatment with saline or MCC-950. (**l**) Caff/Dobu test showing ventricular tachycardia (VT - see red line) in WT + DM treated with saline, but not when treated with MCC-950. (**m**) Arrhythmia score summary after MCC-950 treatment (n = Saline: 6; MCC-950: 4). The results are expressed as mean ± s.e.m. Scatter plot shows values from individual mice, where horizontal bars represent means and error bars, s.e.m. * Represents $P < 0.05$ versus WT + DM (unpaired t-test). # represent $P < 0.05$ (Bonferroni's post test following two-way ANOVA.

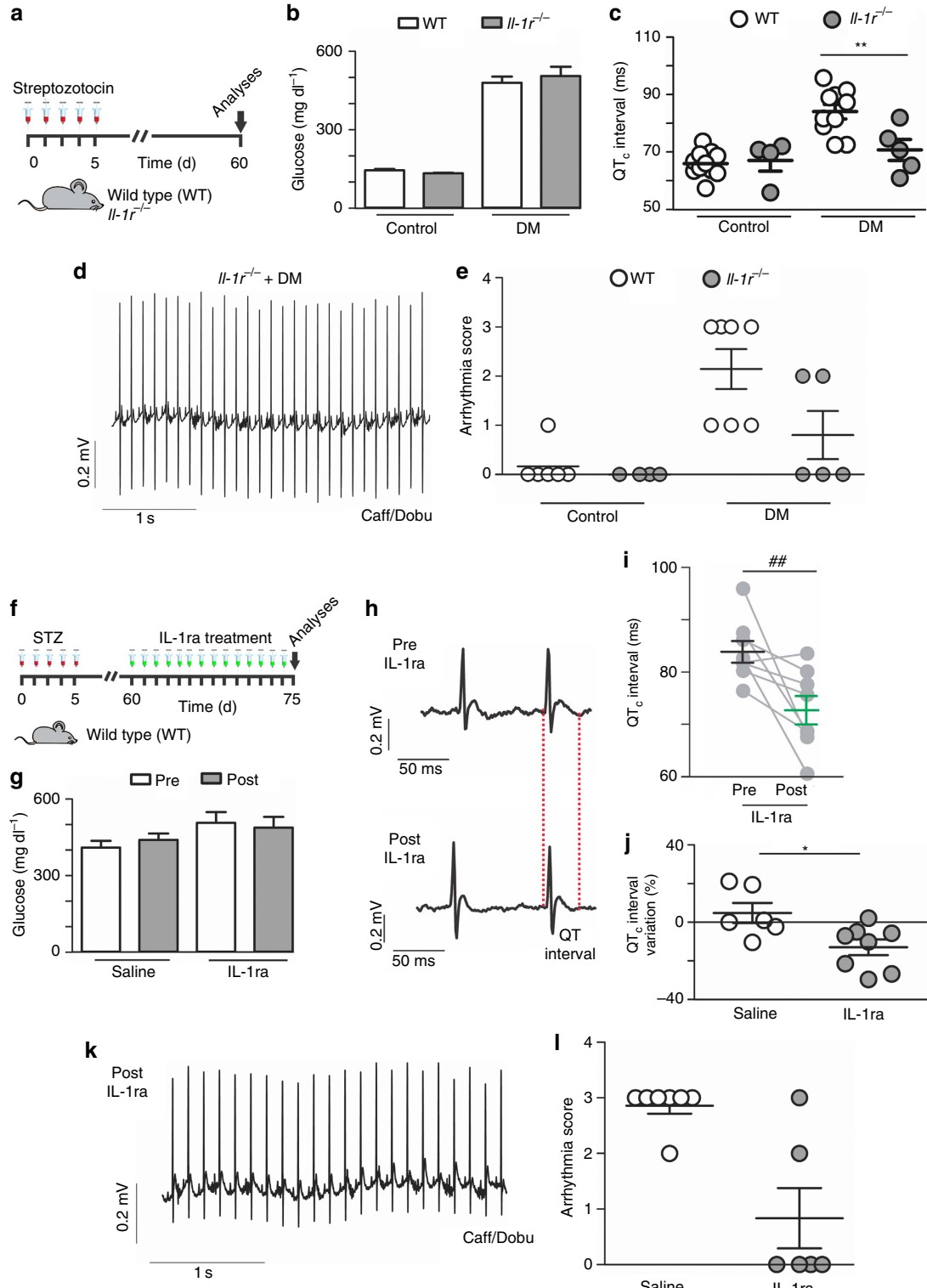

**Figure 7 | IL-1β regulates cardiac electrical remodelling and cardiac arrhythmias.** (**a**) Experimental protocol of diabetes induction in WT and IL-1 receptor *knock-out* mice (*IL-1r⁻/⁻*). (**b**) Fasting blood glucose levels at day 60 (**c**) Corrected QT (QTc) interval duration ($n$ = WT: 10; *Il-1r⁻/⁻*: 4; WT + DM: 10; *Il-1r⁻/⁻* + DM: 5). (**d**) Representative traces of arrhythmic vulnerability test induced by Caff/Dobu showing a normal ECG in *Il-1r⁻/⁻* + DM mice. (**e**) Score quantification of arrhythmia incidence and severity ($n$ = WT: 7; *Il-1r⁻/⁻*: 4; WT + DM: 7; *Il-1r⁻/⁻* + DM: 5). (**f**) Experimental protocol for IL-1ra treatment (Anakinra, 10 mg kg⁻¹ i.p./daily/15 consecutive days). (**g**) Fasting blood glucose levels pre (day 60) and post (day 75) treatment (at least four mice per group). (**h**) Representative ECG traces highlighting QT interval pre and post treatment with saline or IL-1ra. (**i,j**) QTc interval values and their per cent variation pre and post treatment (for control reference please check Fig. 6j). (**k**) Representative ECG traces of WT + DM mice treated with IL-1ra during Caff/Dobu test, showing normal electrical function. (**l**) Arrhythmia score summary after IL-1ra treatment ($n$ = saline: 6; IL-1ra: 8). The results are expressed as mean ± s.e.m. Scatter plot shows values from individual mice, where horizontal bars represent means and error bars, s.e.m. * and ** represents, respectively, $P < 0.05$ and $P < 0.01$ (unpaired $t$-test). ## represent $P < 0.01$ (Bonferroni's *post* test following two-way ANOVA).

oxidation of CaMKII (oxi-CaMKII) and increased CaMKII phosphorylation (p-CaMKII) in cardiac tissue, it is likely that IL-1β acts through CAMKII to promote pro-arrhythmic effects.

QTc prolongation is an independent marker of increased mortality in T1D patients[55,56], and on resting electrocardiograms (ECGs) it is closely associated with susceptibility to tachyarrhythmias. We demonstrate that pharmacological strategies of blocking either the IL-1 receptor or NLRP3 activation were both very effective in reversing QTc prolongation and attenuating the vulnerability to Caff/Dobu induced arrhythmias in our experimental DM model. As IL-1Ra has already been successfully used to improve the control of T1D and T2D[57,58], it could be readily tested in DM patients undergoing IL-1Ra clinical trials to reduce the electrical vulnerability of the heart. Finally, the robust results obtained here using the NLRP3 inhibitor, MCC-950, open a new therapeutic avenue for T1D induced cardiac arrhythmias, since it improves the cardiac electrical profile, while keeping other inflammasomes involved in the response to infectious diseases intact.

## Methods

**Animals and experimental protocol.** This study was carried out in strict accordance with the recommendations of the Guide for the Care and Use of Laboratory Animals of the Brazilian National Council of Animal Experimentation (http://www.cobea.org.br/) and Federal Law 11.794 (October 8, 2008). The protocols of the present study were approved by the Committees for Animal Care and Use at the Federal University of Rio de Janeiro (under no.: 130/13, 131/13 and 102/15), and at the University of Campinas (no. 3658–1). $Tlr2^{-/-}$ male and female mice in C57BL/6a background were kindly provided by Prof. Sergio Costa Oliveira at Federal University of Minas Gerais (Belo Horizonte, Brazil). $Nlrp3^{-/-}$ (Genentech, USA) (# OM-214220) (ref. 17), $Casp1^{-/-}$ (refs 59,60) (MTO # 14865) and $IL-1r^{-/-}$ mice (Jackson, USA) were generated in the C57BL/6 background. Also, male transgenic mice with cardiomyocyte-delimited transgenic expression of either a CaMKII inhibitory peptide (AC3-I) or a scrambled control peptide (AC3-C) were used[32]. Wild-type (WT) C57BL/6 male and female mice were used as controls. Animals, studied at the age of 8–10 weeks, were kept at constant temperature (23 °C) in a standard light/dark cycle (12 h/12 h) with free access to standard chow and water. Animals were divided in four groups: (a) control non-diabetic mice (WT); (b) diabetic mice (WT + DM); (c) non-diabetic transgenic mice ($Tlr2^{-/-}$, $Casp1^{-/-}$, $Nlrp3^{-/-}$, $IL-1r^{-/-}$); and (d) transgenic diabetic mice ($Tlr2^{-/-}$ + DM, $Casp1^{-/-}$ + DM, $NLRP3^{-/-}$ + DM, $IL-1r^{-/-}$ + DM). The mice were bred and maintained under specific pathogen-free conditions at the animal facilities of the Federal University of Rio de Janeiro, and the mice were killed by cervical dislocation. Diabetes was induced by five consecutive injections of (50 mg kg$^{-1}$ i.p.) streptozotocin[61] (Sigma-Aldrich) in 0.05 M citrate buffer (pH 4.5). For in vitro and ex vivo experiments the following compounds were used: TLR2/TLR1 agonist (Pam3CSK4 InvivoGen, USA, (1 μg ml$^{-1}$)), rat recombinant IL-1β (Cat.# 400-01B PeproTech, USA, (10 ng ml$^{-1}$)) human recombinant receptor antagonist (IL-1ra) (PeproTech, USA, (100 ng ml$^{-1}$)). Commercial IL-1ra (Anakirna, Kineret, Swedish Orphan Biovitrum, Sweden)[62] and NLRP3 inhibitor (MCC950, Avistron Chemistry Services, UK)[35]. In some specific experiments 8–12 weeks old male Wistar rats were used.

**Heart-to-body weight and heart-to-tibia length ratios.** Mice were weighted before killing and excised hearts were washed with PBS. Hearts were drained by gentle squeeze using cotton gauze before weighing. Right tibias were gently excised and all muscle was removed in order to expose the whole bone for precise measurements with caliper rule.

**Blood glucose and insulin concentration.** Blood glucose levels were determined using a glucose reagent strip and a standard automated glucometer (AccuChek Advantage II, Roche, Ireland). Briefly, animals were fasted for 5 h and blood was obtained from the tip of the tail vein of the fully awake, non-anesthetized animals. Mice were considered diabetic if fasting glycemia was higher than 300 mg dl$^{-1}$. Insulin levels were measured using a commercial Kit (ImmunoChem Coated Tube—Insulin—MP Biochemicals, USA; Cat.# 07-260102).

**Macrophage depletion.** To deplete macrophages, Clodronate liposomes (Clodronate-L) obtained from Clodronate Liposomes Foundation (The Netherlands) and stored at 4 °C were prepared as a suspension for intravenous (tail vein) injection (10 μl g$^{-1}$) as suggested by the manufacturer and previously described[33,63]. As a control, PBS containing Liposomes (referred as Liposomes) was used at the same final volume.

**Histology and immunohistochemistry.** After weighting the heart was cut transversally, fixed in 4% formaldehyde, and embedded in paraffin. Five micrometre thick sections were obtained and stained with Picrosirius red for collagen content analysis according to standard protocols. Images were obtained with an Axio Zoom.V16 microscope (Zeiss, Germany) and analysed with Image Pro Plus Analyser 7 (Media Cybernetics, USA).

For immunostaining, hearts were dissected, fixed in 2% paraformaldehyde with shaking, cryoprotected in 30% sucrose, and embedded in Tissue-Tek OCT (Miles Laboratories, Pittsburgh, PA, USA). Ten micrometre thick sections were obtained and collected serially onto polyethilenamide-coated glass slides. Before immunolabelling, slides were extensively washed, permeabilized with 0.5% Triton X-100 (Sigma-Aldrich), blocked with 5% bovine serum albumin (Sigma-Aldrich) and finally incubated overnight at 4 °C with primary antibodies. The following antibodies were used: mouse monoclonal anti-troponin T (TnT; 1:100; Thermo Fisher Scientific no. 13-11); goat polyclonal anti-CIAS1/NALP3 (NLRP3; 1:200; Abcam no. ab4207); rabbit polyclonal anti-ASC (ASC; 1:200; Sigma-Aldrich no. SAB4501315), and rat monoclonal anti-F4/80 (F4/80; 1:100; Abcam no. ab6640). After rinsing sections were incubated for 2 h at room temperature with fluorescent dye-conjugated anti-mouse, rat or rabbit antibodies (10 μg ml$^{-1}$; Life Technologies). They were then incubated with 4,6-diamidino-2-phenylindole for 10 min at room temperature. After an additional wash with PBS and one with distilled water, slides were mounted with mounting medium (Prolong). Negative controls were obtained by omitting the primary antibody. Confocal images were obtained in a Nikon A1 confocal laser scanning microscope using Plan Apo lambda 40 × and 60 × objectives for oil immersion (0.6 and 1.49 of numerical aperture, respectively). Images correspond to renderized stacks. Stacks (images captured 0.3 μm apart in Z-plane) of XY images were collected, and to avoid bleed-through between channels, images at each wavelength were captured sequentially. Images were assembled, prepared for analysis uniformly and XY and XZ images were generated using ImageJ software (NIH, USA). To reduce variability among samples sections from different groups were processed simultaneously for each marker and the setting for image acquisition was the same for all the images. The fluorescence area (anti-NALP3) was measured using the ImageJ software (NIH, USA).

**Flow cytometry.** Hearts were thoroughly perfused with PBS, minced into 1 mm$^3$ fragments and digested with 1 mg ml$^{-1}$ collagenase II (Worthington) in DMEM. Fragments were submitted to 5–6 cycles of digestion under gentle agitation at 37 °C. The supernatant was collected after every cycle and stored in DMEM with 10% FBS. Subsequently, samples were filtered through a 100 μm strainer, washed with 0.5% bovine sera-albumin (BSA) in PBS and submitted to Fc receptor blocking with Mouse BD Fc Block (BD Biosciences, Cat# 553141) according to the manufacturer's instructions. The following primary antibodies were added directly to the blocked samples and incubated for 30 min at 4 °C: CD45-PerCP (BD Biosciences, Cat# 557235 1:100), CD11b-FITC (BD Biosciences, Cat# 553310 1:100), F4/80-PE (BD Biosciences, Cat# 563899 1:50), Ly6C-APC (BioLegend, Cat# 128003 1:25) and MHCII-PE-Cy7 (BioLegend, Cat# 116419 1:100). Samples were washed and incubated with 0.25 μg ml$^{-1}$ 4,6-diamidino-2-phenylindole for 5 min at room temperature. Data were acquired in BD FACSAriaIIu. The gating strategy was as follows: CD45$^+$ → doublet discrimination (FSC-H, FSC-W) → dead cell exclusion → CD11b$^+$ F4/80$^+$ → morphology (SSC-A, FSC-A) → MHCII$^{high}$Ly6C$^-$ (Supplementary Fig. 8). Data were analysed using FlowJo v10 software.

**Calcium transient and cell shortening measurement.** Myocyte Isolation: rats (200–350 g) were anaesthetized by isoflurane inhalation or an intra-peritoneal injection of sodium pentobarbitone (35 mg kg$^{-1}$ body weight,), and hearts were rapidly excised. Cardiac myocytes were isolated with collagenase digestion using a previously described technique[64]. Briefly, the hearts were attached via the aorta to a cannula, excised and mounted in a Langendorff apparatus. They were then retrogradly perfused at 37 °C at a constant perfusion pressure of 70–80 mm Hg with Krebs-Henseleit solution (K-H) of the following composition (mM): 146.2 NaCl, 4.7 KCl, 1.00 CaCl$_2$, 10.0 N-2-hydroxyethylpiperazine-N'-2-ethanesulfonic acid (HEPES), 0.35 NaH$_2$PO$_4$, 1.05 MgSO$_4$ 10.0 glucose (pH adjusted to 7.4 with NaOH). The solution was continuously bubbled with 100% O$_2$. After a stabilization period of 4 min, the perfusion was switched to a nominally Ca$^{2+}$-free K-H for 6 min. Hearts were then recirculated with type 2 collagenase (118 units ml$^{-1}$), 0.1 mg ml$^{-1}$ protease and 1% BSA, in K-H containing 50 μM CaCl$_2$. Perfusion continued until hearts became flaccid (10–15 min). Hearts were then removed from the perfusion apparatus by cutting at the atria-ventricular junction. The isolated myocytes were separated from the undigested tissue and rinsed several times with a K-H solution containing 1% BSA and 500 μM CaCl$_2$. After each wash, myocytes were left for sedimentation during 10 min. Myocytes were kept in K-H solution at room temperature (20–22 °C) until use. After isolation, cells were incubated for 24 h at 4 °C in modified KB solution (containing: 70 mM potassium gluconate, 10 mM KCl, 20 mM taurine, 1 mM MgCl$_2$, 10 mM glucose, 10 mM HEPES; 0.3 mM EGTA; pH 7.4) in the presence or absence of 10 ng ml$^{-1}$ IL-1β and/or 2.5 μM of the CaMKII inhibitor KN93. EGTA was omitted from the solution in the experiments in which the cytosolic Ca$^{2+}$ concentration ([Ca$^{2+}$]i) was quantitated. Cells from the same heart were used for incubation with and without IL-1β (10 ng ml$^{-1}$) for 24 h at 4 °C. For ([Ca$^{2+}$]i) measurement, myocytes were loaded with 10 μM indo-1 AM (Molecular Probes, USA) during 15 min at room

temperature. The perfusion chamber containing the cells was placed on the stage of an inverted microscope (Nikon Diaphot 200) coupled to a microfluorimetry system (Photon Technology Intl, Inc.). The dye was excited at 360 nm, and the emission ratio at 410 and 485 nm was converted to $[Ca^{2+}]i$ according to Bassani et al.[64], using calibration parameters determined in live cells. Myocyte contractile activity was evaluated simultaneously by monitoring peak cell shortening (expressed as per cent of resting cell length) under electrical stimulation at 1 Hz. The propensity to develop arrhythmias was estimated from the number of non-stimulated contractile events (NSE), which were defined as the spontaneous contractions developed after interruption of a 10 min-long stimulation train at either 0.5 or 3 Hz. To investigate the role of a possible increase in $Ca^{2+}$ extrusion rate via the $Na^+/Ca^{2+}$ exchanger (NCX) as a factor that may facilitate the generation of $Ca^{2+}$-dependent spontaneous electrical activity, the ratio of the rate constants of $[Ca^{2+}]i$ decay at a caffeine-evoked transient and a twitch (in which cytosolic $[Ca^{2+}]i$ removal is mainly attributable to the NCX and the SR $Ca^{2+}$-ATPase, respectively), was compared in control and IL-1-treated myocytes[64]. To estimate the rest-dependent loss of the SR $Ca^{2+}$ content, which may occur in case of increased rate of diastolic SR $Ca^{2+}$ release[25], the content of $Ca^{2+}$ stored in the SR (as ⌈moles $Ca^{2+}$ per liter of non-mitochondrial cell water) at steady-state was estimated from a $Ca^{2+}$ transient evoked by caffeine in $Na^+$-, $Ca^{2+}$-free solution[26] after stimulation at 1 Hz for 3 min. The protocol was repeated with interposition of a 2 min-long rest period before caffeine application. The variation in SR $Ca^{2+}$ load per min of rest was calculated based on the difference of the load in the steady-state (that is, non-rested) and post-rest conditions. For detection of spontaneous $Ca^{2+}$ release myocytes were loaded with 10 μM of Fluo-3 AM (Molecular Probes) and mounted on the stage of a Zeiss 410 inverted confocal microscope (LSMTech, USA). Cells were imaged in linescan mode along their long axis, with excitation via the 488 nm line of an argon laser emission was collected at >515 nm. Each image consisted of 512 line scans obtained at 4 ms intervals. Data were analysed using the 'Sparkmaster' plugin for ImageJ (NIH, USA). Sparks were imaged in quiescent cells after 10 min stimulation at either 0.5 or 3 Hz.

**Western blot.** Left ventricle tissue samples were incubated for 24 h in the presence or absence of $10 \, ng \, ml^{-1}$ IL-1β and subsequently homogenized. Protein was measured by the Bradford method using BSA as standard. Lysates (~90 μg of total protein per gel line) were seeded in a 10% SDS polyacrylamide gel and transferred to polyvinylidene difluoride membranes. Blots were probed overnight with antibodies raised against oxidized CaMKII (referred as Ox-CaMKII), Cat# 07-1387, 1:1,000 (Millipore Corp, USA). After stripping the blots were probed with phospho-Thr286-CaMKII (referred as p-CaMKII), Cat# 32678 1:1,000 (Badrilla, UK), Anti IL-1β Cat# I3767 1:1,000 (Sigma, USA) and Anti GAPDH, Cat# MAB374, 1:2,000 (Santa Cruz Biotechnology, USA) was used for normalization. Immunoreactivity was visualized by a peroxidase-based chemiluminescence detection kit (Amersham Biosciences) using a Chemidoc Imaging System. The signal intensity of the bands in the immunoblots was quantified by densitometry using the Image J software (NIH, USA).

**Magnetic resonance images.** *In vivo* magnetic resonance cardiac imaging (MRI) was performed on a 7.0 T horizontal-bore MR scanner (Varian Inc, USA) under inhalation anesthesia via a nose cone (isoflurane 1.5 vol% supplemented by 0.5 liter oxygen per minute). Rectal temperature was monitored, and body temperature was kept constant at 36.5 °C through an air heating system. The ECG was recorded at two precordial subcutaneous electrodes leads and respiration was monitored with a pneumatic pillow. Measurements were transmitted through a monitoring and gating system (model 1025 Monitoring and Gating System; SA Instruments, USA) to the MRI system. High-resolution bright-blood MRI experiments were conducted using an ECG-triggered fast spoiled gradient echo cine sequence in a number of adjacent short- and long-axis slices. The scanning parameters were optimized for the signal-to-noise ratio as follows: flip angle = 30°, echo time (TE) = 1.9 ms, repetition time (TR) ≅ R-R interval, RF pulse width = 1.0 ms, number of averages = 8; 15 frames per heart cycle were obtained. All images were acquired with a field of view of $30 \times 30$ mm and a data matrix of $128 \times 128$ mm to yield an in-plane resolution of 234 μm². Total scan time was in the range of 25 min. Each imaging protocol resulted in five to seven 1 mm thick short-axis images covering the whole heart from apex to base with no gap between slices. The data were analysed with Osirix Imaging software. Ventricular slice volumes were determined from end-diastolic and end-systolic images by multiplication of the compartment area and slice thickness. Total volumes were calculated as the sum of the volume of all slices and the ejection fraction was calculated by Simpson's rule.

**ECG and arrhythmia incidence/severity test.** ECG recording was carried out in conscious animals with a noninvasive method[65]. Electrodes were positioned in the DI lead and connected by flexible cables to a differential AC amplifier (model 1700, A-M Systems, USA), with signal low-pass filtered at 500 Hz and digitized at 1 kHz by a 16-bit A/D converter (Minidigi 1-D, Axon Instruments, USA) using Axoscope 9.0 software (Axon Instruments, USA). Data were stored for offline processing. For the analysis of percentage variation of the QTc interval, the following calculation was applied: ((Vf-Vi)/Vi × 100) where Vf = Final value (after the treatment) and Vi = Initial value (before the treatment).

To perform the arrhythmia vulnerability test, anesthetized animals (Xilazine—Syntec, Brazil, $(0.5 \, mg \, g^{-1})$) and ketamine—Cristália, Brazil, $(1.5 \, mg \, g^{-1})$) received an injection of caffeine $(120 \, mg \, kg^{-1}$, i.p.) and dobutamine $(50 \, \mu g \, kg^{-1}$, intravenous)[29]. A basal ECG was recorded for 3 min before caffeine/dobutamine application. After the treatment, the ECG was recorded for 15 min. An adapted arrhythmic score of incidence and severity was applied to each animal[29,66]: 0 if there were no arrhythmic events during the test; 1, if one premature ventricular contractions were present; 2, if bigeminy and/or salvos were present; 3, for ventricular tachycardia; 4, in case of ventricular fibrillation; and 5, in case of spontaneously induced ventricular fibrillation.

**Cardiac action potential recording.** APs were recorded from murine left ventricular muscle strips. The cardiac strips were immobilized at the bottom of a tissue dish exposing the endocardium[67]. The preparations were superfused with Tyrode's solution containing (in mM): 150.8 NaCl, 5.4 KCl, 1.8 $CaCl_2$, 1.0 $MgCl_2$, 11.0 D-glucose, 10.0 HEPES (pH 7.4 adjusted with NaOH; 37.0 ± 0.5 °C) saturated with oxygen at a perfusion flow rate of $5 \, ml \, min^{-1}$ (Miniplus 3, Gilson, France). The tissue was stimulated at three different basic cycle lengths (200, 300 and 500 ms). Transmembrane potential was recorded using glass microelectrodes (10–40 MΩ DC resistance) filled with 2.7 M KCl connected to a high input impedance microelectrode amplifier (Electro 705, World Precision Instruments, USA). Amplified signals were digitized (1,440 digidata A/D interface, Axon Instrument, Inc.) and stored in a computer for later analysis using LabChart 7.3 software (ADInstruments, Australia). The following parameters were analysed: resting membrane potential, action potential amplitude and APD at 90% (APD₉₀) repolarization. For specific set of experiments left ventricular muscle strips, from Wistar rats were incubated for 24 h at 4 °C with Tyrode's solution containing agonist or not as previously described[68].

**Transient outward potassium current ($I_{to}$) recording.** Immediately after isolation, rats cardiomyocytes were suspended in modified KB solution containing human recombinant IL-1β (PeproTech, USA, $(60 \, pg \, ml^{-1})$) or not (control group) and incubated in this solution at 4 °C for 24 h. Cells were transferred to a shallow chamber and allowed to settle for at least 10 min before superfusion with the external bathing solution. For the experiments we used only $Ca^{2+}$-tolerant rod-shaped cells, with clear cross-striations and lacking any visible blebs on their surface. All experiments were performed at room temperature (20–22 °C). Ion currents were recorded using the whole-cell configuration of the patch-clamp technique with an Axopatch 200B amplifier (Axon Instruments Inc., USA). Recording pipettes were obtained from borosilicate tubes (Sutter Instruments, USA), and had a tip resistance of 1–3 MΩ when filled with the internal solution (in mmol $l^{-1}$): L-aspartic acid (potassium salt) 80, $KH_2PO_4$ 10, $MgSO_4$ 1, KCl 50, HEPES-K⁺ 3, ATP-Na₂ 3, EGTA-K⁺ 10, pH adjusted to 7.2 with KOH. Following the patch rupture, whole-cell membrane capacitance was measured by integration of the capacitive transients elicited by voltage steps from −50 to −60 mV, which did not activate any time-dependent membrane current. Series resistance was compensated by 80% in order to minimize voltage errors, and checked regularly throughout the experiment. The voltage-clamp experimental protocols were controlled with the 'Clampex' program of the 'pClamp 10.2' software (Axon Instruments Inc., USA). The external bathing solution was (in mmol $l^{-1}$): NaCl 86, $MgCl_2$ 1, HEPES-Na⁺ 10, KCl 4, $CaCl_2$ 0.5, $CoCl_2$ 2, dextrose 12, TEA-Cl 50, pH adjusted to 7.4 with NaOH. $I_{to}$ was recorded during depolarizing pulses from - 30 mV to + 50 mV, starting from a holding potential of -60 mV (ref. 68). Pulses were applied at a frequency of 0.1 Hz to ensure the full recovery of $I_{to}$ from inactivation. The TEA-resistant time-independent $I_{ss}$ was digitally subtracted. $I_{to}$, considered as the peak current $I_{to}$ minus $I_{ss}$, was normalized to cell capacitance, and expressed as pA/pF. Steady-state inactivation was determined using a two pulse protocol consisting of a 5 s conditioning prepulse ranging from −90 to +10 mV in 10 mV increments, followed the 500 ms test pulse to +50 mV. Peak current amplitude at +50 mV was plotted against the prepulse voltage. To study the time-dependent recovery of $I_{to}$, a 150 ms conditioning pre-pulse to +50 mV to induce channel inactivation was followed by a test pulse to +50 mV, applied at increasing intervals.

**Microelectrode array analysis (MEA).** MEA measurements in cardiomyocytes derived from induced pluripotent stem cells (iPS) (Cor.4U—Axiogenesis, Germany) were performed [69]. The cells were tested to mycoplasma detection before use. Cardiomyocytes were detached with 0.05% Trypsin in 0.5 mM EDTA (Gibco/Life Technologies) for 5 min at 37 °C, centrifuged for 5 min at 1,000 r.p.m. and resuspended in Cor.4U Culture Medium. Then, 20,000 cells were plated in each well of a 6-well MEA chamber (60-6wellMEA200/30iR-Ti-tcr, Multi-Channel Systems, Germany) coated with 0.01% fibronectin (Sigma-Aldrich). After 72 h, the Cor.4U Culture Medium was replaced with Tyrode's solution and field potentials were recorded at a sampling rate of 10 kHz with the MC-Rack software at 37 °C. The FPD was manually measured from the minimum of the sharp negative spike to the following maximum. For analysis of the per cent variation of FPD the following calculation was applied: ((Vf-Vi)/Vi × 100) where Vf = Final value (after incubation with IL-1β, IL-1 + AIP or vehicle) and Vi = Initial value (before incubation).

**Cytokine level measurements.** Il-1β levels were measured using an enzyme-linked immunosorbent assay kit according to manufacturer's instructions (R&D systems, USA)[70].

**Statistical analysis.** For statistical analysis, the following tests (two-tailed) were applied: Student's t-test for comparisons between two groups in the case of one independent factor; one-way analysis of variance (ANOVA) in the case of one independent factor, when comparing more than two groups; two-way ANOVA in the case of two independent factors, followed by the Bonferroni test for multiple comparisons; in case of not normally distributed data, either Mann–Whitney test or Kruskal–Wallis test followed by Dunn's multiple comparisons was applied. $P < 0.05$ values were considered statistically significant. Data are presented as mean ± standar error of the mean (s.e.m.). All analyses were made using GraphPad Prism 5.0 (GraphPad Software, USA). We did not use statistical methods to predetermine sample size, there was no randomization designed in the experiments, and the studies were not blinded. Samples sizes were estimated on the basis of sample availability and previous experimental studies of the cardiovascular system[9–13,16].

**Data availability.** The authors declare that the data supporting the findings of this study are available within the article and its Supplementary Information Files or from the corresponding authors upon reasonable request.

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

## Acknowledgements

We are grateful to Dr Dario Zamboni from University of São Paulo for supplying $Il-1r^{-/-}$ mice and to Dr Richard Flavell from Yale University Medical School who allowed us to use the $Casp1^{-/-}$ mice. In addition, we would like to thank Prof. Sergio Costa Oliveira at Federal University of Minas Gerais (Belo Horizonte, Brazil) for providing $Tlr2^{-/-}$ mice. We also wish to thank Dr Mark Anderson from Johns Hopkins University School of Medicine—USA, for providing AC3-I/C mice breeding pairs to 'Centro de Investigaciones Cardiovasculares—UNLP—Argentina'. We would like to thank Maximilian Funken from University Bonn for assistance with microelectrode array experiments. We thank Axiogenesis AG for providing hiPS-derived cardiomyocytes and Genentech, which authorized the use $Nlrp3^{-/-}$ mice. We also thank Drs Tobias Bruegmann, Eicke Latz and Philipp Sasse from University of Bonn for scientific discussions, Dr Fernando Pereira de Almeida from Federal University of Rio de Janeiro for assistance with microscopy image acquisition, Sandro Torrentes da Cunha from Federal University of Rio de Janeiro for assistance with histology preparations and Dr Bernardo Tura from Brazilian National Institute of Cardiology for assistance with the statistical analyses. Also we would thank Isis Hara Trevenzoli from UFRJ for help us to perform the insulin assay. We thank Mónica Rando and Omar Castillo from Centro de Investigaciones Cardiovasculares - UNLP for technical support.

This work was funded by the Brazilian National Research Council (CNPq, grants: 308168/2012-7 and 475218/2012-4), the Carlos Chagas Filho Rio de Janeiro State Research Foundation (FAPERJ, grants: E-26/103.222/2011 and E-26/111.171/2011) and National Institutes of Science and Technology for Biology Structural and Bioimaging (grant: 573767/ 2008-4), Brazil and by grants of the Deutsche Forschungsgemeinschaft (FL 276/7-2 to B.K.F.) and by the Stem Cell Factory II co-founded by the European Union (European Regional Development Fund—Investing in your future) and the German federal state North Rhine-Westphalia (NRW) (to D.M. and B.K.F.). Additionally, the work was funded by PICT 1678 from FONCYT to M.V.-P.. G.M. has a postdoctoral fellowship from FAPERJ (PDR 10), F.F.D has a postdoctoral fellowship from CAPES, L.R.V. has a PhD fellowship from FAPERJ, M.L.A has a PhD fellowship from CNPq.

## Author contributions

G.M., M.L.A., E.M., designed the study. G.M., M.L.A., L.R.V., C.H.-M., G.B., R.A.B., O.C., A.B.C., J.I.B., F.F.D., D.M., M.S., performed experiments. G.M., M.L.A., L.R.V., C.H.-M., G.B., R.A.B., O.C., A.B.C., J.I.B., F.F.D., D.M., L.H.T., M.S., M.V.-P., M.T.B., C.N.P., A.B., B.K.F., A.C.C.C., E.M., performed data analysis and interpretation. G.M., C.P., B.K.F., A.C.C.C., E.M., wrote the paper with input from the other authors.
