## [Peer Review File · Nature Communications]

Reviewers' comments:

Reviewer #1 (expert in diabetic cardiomyopathy)

Remarks to the Author:

In this study, the authors demonstrated that the activation of inflammasome and induction of IL-1b is responsible for the DM associated arrhythmia. The authors need to address the following Major Concerns:

1. It has been reported that cytokines, including IL-1b, influence the electrical and contractile functions of cardiac myocytes. The novelty of this manuscript is studying the function of IL-1b in the setting of DM induced arrhythmia. Therefore, the underline mechanism of the induction of myocardial inflammation (specifically, the increased expression of IL-1b) by DM is better to be addressed not only in discussion, but also by some data.
2. Insulin signaling plays an important role in cardiac electrophysiology. It has been shown that impaired insulin signaling causes abnormal repolarization in cardiomyocytes. What is the insulin level in the diabetic model generated in this study? Are the insulin levels in the WT and Tlr2^{-/-}, NLRP3^{-/-}, Casp1^{-/-}, and IL-1r^{-/-} DM mice comparable?
3. According to the results (Figure 1h), the serum IL-1b level is around 60pg/mL in WT DM mice. However, the isolated cardiac myocytes were treated with IL-1b at 10ng/mL, which is more than 100 times of the pathophysiological level. The experiments have to be performed using a more pathophysiological relevant dose of IL-1b.
4. The animal models used in this study are all global KOs. The attenuation of inflammation may provide systemic protection, such as protecting pancreatic beta cells against apoptosis. The authors also mentioned in the discussion that the treatment with IL-1Ra can reduce insulin dependence in the T1D children. Thus, the insulin levels in these KO DM mice may be different from their WT controls. Again, insulin signaling influences the electrophysiological response of cardiac myocytes. How to distinguish the effects of global attenuation in inflammation in the KO mice from the local effects of anti-arrhythmic in the myocardium?
5. It has been reported that IL-1b reduces L type Ca²⁺ channel current density in cardiomyocytes. (J Biol Chem. 2014, 289(32):21896-908.) How is the I_{CaL} in the models tested in this study?

Minor points:

1. What is the expression level of Tlr2 in the wild type mice? Does it increase in the DM WT mice?
2. How about the expression level of other cytokines, such as TNF α ?

Reviewer #2 (expert in inflammasome and diabetic cardiomyopathy)

Remarks to the Author:

Monnerat et al present a manuscript exploring the role of IL-1b in diabetes induced cardiac arrhythmias in mice. They show that macrophages expressing TLR2 sense endogenous ligands triggering the expression of the NLRP3 inflammasome and IL-1b. Through unknown signals the NLRP3 inflammasome is activated causing release of IL-1b. It appears that IL-1b in its own right promotes the arrhythmia. Using genetically deficient mice they prove the importance of this pathway. Finally they show the translatable ability of their findings as administration of Anakinra or MCC950 could reduce diabetes induced arrhythmias.

Comments:

1. Figure 1 - the authors went directly to look at TLR2 but it is unclear why TLR2 and not TLR4 or

RAGE were not also considered? While the TR2 KO mice are clearly protected, there should be a better rationale why this was examined only. Also is TLR2 unregulated on cardiac macrophages from diabetic animals?

2. Figure 1i: The western blot is difficult to assess can the quality be improved? Can you also perform gene expression to compliment this result?

3. In all of these genetic and pharmacological interventions was there any changes in heart weight or hypertrophy?

4. Figure 5J and Figure 6I: There appears to be a replication of the data in these two figures - the control groups are the same for both. If these interventions share the same control group they should be graphed in the same figure.

Reviewer #3 (expert in cardiac arrhythmia)

Remarks to the Author:

This is an interesting and potentially important paper that provides convincing evidence for an association between hyperglycemia, activation of IL-1B sourced from macrophages, and arrhythmias. The strengths of this study are the insights into the underexplored connections between non-myocyte inflammatory cells and signaling and myocardial disease and the clinical/translational implications of these findings. The weaknesses are the failure to 'connect the dots' between various phenomena: is loss of Ito a critical step in the arrhythmias? If so it is likely less relevant to larger mammals (including humans) because Ito is far more important in rodents where Ito is critical for their abbreviated AP duration. Is AP prolongation necessary to activate CaMKII or does increased oxidation and/or O-GlcNAcylation activate CaMKII? Although CaMKII appears essential to the proposed arrhythmogenic pathway, its role is only explored in cursory fashion (e.g. in rodent and not iPS cells and only using KN-93). KN-93 has numerous off target actions, including ionic currents with potential relevance to the observed arrhythmia phenotypes. In contrast, these off target actions are not shared by the 'control' compound KN-92. Thus, cleaner approaches to CaMKII inhibition are required. Fortunately, many options are available including mice lacking CaMKII δ , mice with myocardial transgenic expression of AC3-I (I believe these are in the laboratory of the nearby La Plata group), mice with oxidation-resistant CaMKII, shRNA knock down for cellular studies, etc.

Reviewers' comments:

Reviewer #1 (expert in diabetic cardiomyopathy)

Remarks to the Author:

In this study, the authors demonstrated that the activation of inflammasome and induction of IL-1b is responsible for the DM associated arrhythmia. The authors need to address the following:

Major Concerns:

1. It has been reported that cytokines, including IL-1b, influence the electrical and contractile functions of cardiac myocytes. The novelty of this manuscript is studying the function of IL-1b in the setting of DM induced arrhythmia. Therefore, the underline mechanism of the induction of myocardial inflammation (specifically, the increased expression of IL-1b) by DM is better to be addressed not only in discussion, but also by some data.

Answer: The effects of IL-1 β on the electrical function of cardiomyocytes reported in the literature are still elusive. A recent review¹ on this subject highlights the rather indirect evidences linking IL-1, TNF and IL-6 to delayed repolarization, long QT syndrome and ventricular tachycardia. In 1993, IL-1 (not the mature IL-1 β) was shown to increase the duration of the action potential², and this is apparently the only direct evidence of its action on electrical function of cardiomyocytes. Another study described acute effects of IL-1 β on atrial preparations, and did not find any effects on action potential duration³. Thus, studies showing IL-1 effects on action potential duration, ventricular tachycardia or long QT syndrome are scant. For diabetes or even any other disease we provide to the best of our knowledge the first evidence of direct IL-1 β effects on the heart, as follows:

- 1- IL-1 β acting directly through its receptor is able to induce a prolongation of the action potential duration (Fig. 2 b,c) via reducing the I_{to} current (Fig. 2 e,f);
- 2- Cardiomyocytes incubated 24 h in the presence of IL-1 β showed higher frequency and amplitude of cardiac calcium sparks (Fig. 2 g-j); these effects have the potential to induce cardiac arrhythmias;
- 3- IL-1 β induced CaMKII oxidation and phosphorylation in cardiomyocytes, promote a higher incidence of NSE (Fig.3 a-d);
- 4- IL-1 β prolongs also in human cardiomyocytes (hIPS-CM) field potential duration, in this response CaMKII activation is involved (Fig. 2 k-m and Fig. 3 e,f).

Thus, we therefore consider the data provided novel and relevant, as these illustrate IL-1 β -mediated direct modulation of the electrophysiological activity of ventricular cardiomyocytes.

The following text passages have been added to the discussion section of the revised manuscript: (Page: 8; Line: 324)

The direct effects of IL-1 β on electrical function of cardiomyocytes reported in the literature are scant. A recent review¹ on this subject highlights the rather indirect evidences linking IL-1, TNF and IL-6 to delayed repolarization, long QT syndrome and ventricular tachycardia. In 1993, IL-1 (not the mature IL-1 β) was shown to increase the duration of the action potential². Another study described acute effects of IL-1 β on atrial preparations, and did not find any effects on action potential duration³. Herein we demonstrate that IL-1 β creates a pro-arrhythmic environment both in rodent and human cells by reducing the repolarizing K⁺ current (I_{to}), by increasing CaMKII oxidation/ phosphorylation and Ca²⁺ spark frequency. A prolongation of the cardiac AP due to lower I_{to} current has been described in diabetic experimental models^{46,27}, and in agreement with these data, we show here that binding of IL-1 β to its receptor in the heart can produce similar effects. Hyperglycemia was previously reported to enhance CaMKII-dependent activation of spontaneous SR Ca²⁺ release events^{21,23}. We also provide evidence that IL-1 β increases both Ca²⁺ spark frequency and loss of SR Ca²⁺ content during rest in cardiac cells, indicating enhanced SR Ca²⁺ leak. Moreover, although the possibility that IL-1 β could promote oxidation and activation of CaMKII has not been explored in cardiac tissue, recent evidence suggests that this could be achieved via MyD88 inflammatory signaling²². Since the IL-1 β receptor uses MyD88 as a signaling adaptor and we illustrate that IL-1 β promotes oxidation of CaMKII (oxi-CaMKII) and increased CaMKII phosphorylation (p-CaMKII) in cardiac tissue, it is likely that IL-1 β acts through CAMKII to promote pro-arrhythmic effects.

As for the underlying mechanism of cardiac inflammation, we show that macrophage activation (by Pam3) in cardiac strips is able to induce higher levels of IL-1 β secretion leading to prolongation of the cardiac action potential via activation of the IL-1 β receptor (Fig. 4 a-d). These results are corroborated by experiments in which cardiac strips are depleted of macrophages (by clodronate) and treated with Pam3; these cells don't show anymore increased action potential duration (Fig. 4 f,g).

In regard to the specific mechanism involving in cardiac inflammation in diabetes, our results reveal that the MHC^{high} macrophage sub-population, which was previously shown to have greater NLRP3 content and consequently greater IL-1 β secretion⁴, was smaller in hearts from *Tlr2*^{-/-} diabetic mice, than in those from WT-DM mice (Fig. 5 c). In line with this result, the absence of TLR2 prevented upregulation of NLRP3 content in heart macrophages in response to diabetes (Fig. 5 d, e). Another important finding is that lack of NLRP3 receptor or caspase 1 is sufficient to prevent the diabetes-induced ventricular arrhythmias (Fig. 6 g). Interestingly, similar results are obtained, when inhibiting NLRP3 by MCC-950 (Fig. 6 m). Taken together, these data reinforce the idea that the TLR2/NLRP3/ IL-1 β axis is involved in diabetes-induced cardiac inflammation.

*2. Insulin signaling plays an important role in cardiac electrophysiology. It has been shown that impaired insulin signaling causes abnormal repolarization in cardiomyocytes. What is the insulin level in the diabetic model generated in this study? Are the insulin levels in the WT and *Tlr2*^{-/-}, *NLRP3*^{-/-}, *Casp1*^{-/-}, and *IL-1r*^{-/-}*

DM mice comparable?

Answer: We thank the reviewer for this very helpful and important suggestion. We have performed new experiments and measured the circulating insulin levels in WT and all the *knockout* strains used in the manuscript. Our results show that WT diabetic mice present lower insulin levels than WT non-diabetic mice (Supplementary Table 1); similar results were observed in the *Il-1r^{-/-}* diabetic and non-diabetic mice (new Supplementary Table 3). We observed a trend towards lower insulin levels in *Tlr2^{-/-}* and in *Casp1^{-/-}* mice, when comparing diabetic to non-diabetic mice, even though the values did not reach statistical significance ($P = 0.07$ and $P = 0.057$, respectively; Supplementary Tables 1 and 2).

In contrast, the *Nlrp3^{-/-}* mice showed similar circulating insulin levels, when comparing diabetic to non-diabetic mice. In addition, insulin levels in *Nlrp3^{-/-}* mice were similar to those found in WT non-diabetic mice (new Supplementary Tables 1 and 2). Despite the differences observed in the insulin levels, none of the diabetic mice had normal glycemc levels.

These new data are in line with previous results published by Dasu et al.⁵ using the same DM model, namely decreased circulating levels of insulin in both WT and *Tlr2^{-/-}* diabetic mice compared to the respective non-diabetic controls.

Because abrogation of the prolonged QTc interval independent of the respective insulin levels is a constant finding in all mice, we conclude that these cannot underlie the observed electrophysiological alterations.

We have integrated the new data and modified the manuscript text as follows:

i- In a similar manner compared to an earlier study⁵, we have observed a trend towards lower insulin levels in *Tlr2^{-/-}* diabetic compared with *Tlr2^{-/-}* non-diabetic mice ($P = 0.07$) (Supplementary Table 1. In addition, the *Tlr2^{-/-}* diabetic mice showed insulin levels similar to WT+DM mice suggesting that: i - the anti-inflammatory protection afforded by the lack of TLR2 was not enough to protect the streptozotocin-impaired pancreatic function; and ii- the lack of electrophysiological changes observed in this *KO* mice was un-related to changes in insulin levels (page: 4; line: 99).

ii- *Nlrp3^{-/-}* diabetic mice showed insulin levels similar to *Nlrp3^{-/-}* non-diabetic mice ($P = 0.17$), despite their increased glycemia levels. However, *Casp1^{-/-}* diabetic mice presented a trend towards lower levels of insulin, than *Casp1^{-/-}* non-diabetic mice ($P = 0.057$; Supplementary Table 2). These results suggest that while the electrophysiological improvement observed could not be related to the rescue of the insulin levels in the *Casp1^{-/-}* diabetic mice, we can not exclude that insulin could be also involved in the electrophysiological improvement observed in *Nlrp3^{-/-}* diabetic mice (page: 6; line: 226).

iii- *Il-1r^{-/-}* developed hyperglycemia and lower insulin levels in response to streptozotocin ($P = 0.0001$ and $P = 0.04$ respectively; Supplementary Table 3;

Fig. 6b) (page: 7; line: 253).

In addition, we have added the following paragraph to the discussion (Page: 8; Line: 306)

The absence of insulin signaling can reduce I_{to} current, resulting in AP and QT prolongation⁶. Nevertheless, the circulating insulin levels in diabetic *Il-1r^{-/-}* mice were significantly decreased compared to non-diabetic counterparts, while a strong trend towards decreased levels was found in diabetic *Tlr2^{-/-}* and *Casp1^{-/-}* mice. Only diabetic *Nlrp3^{-/-}* mice presented circulating insulin levels similar to non-diabetic controls. Although we cannot exclude that preserved circulating insulin levels account for the prevention of electrophysiological abnormalities in diabetic *Nlrp3^{-/-}* mice, this is certainly not the case in *Il-1r^{-/-}*, *Tlr2^{-/-}* and *Casp1^{-/-}* mice.

3. According to the results (Figure 1h), the serum IL-1 β level is around 60pg/mL in WT DM mice. However, the isolated cardiac myocytes were treated with IL-1 β at 10ng/mL, which is more than 100 times of the pathophysiological level. The experiments have to be performed using a more pathophysiological relevant dose of IL-1 β .

Answer: We thank the reviewer for raising this important point and have therefore performed new experiments using more physiological IL-1 β concentrations: We show a new I/V relationship curve for I_{to} (Fig. 2f) obtained from isolated cardiomyocytes incubated for 24 h with 60 pg/mL of IL-1 β (as shown in Fig. 1h for diabetic wild-type mice). As can be seen in Fig. 2F, under these conditions a 35% reduction in I_{to} current could be observed, demonstrating the sensitivity of this particular ion channel to this cytokine. In addition, we have modified Fig. 2f and the description of the data in the revised results sections as follows (page: 4; line: 122).

Incubation of isolated ventricular cardiomyocytes in the presence of IL-1 β , using pathophysiological levels (60 pg/mL, for more details please see Fig. 1h), for 24 h resulted in an approximate 35 % reduction of this current (Fig. 2d-f).

4. The animal models used in this study are all global KOs. The attenuation of inflammation may provide systemic protection, such as protecting pancreatic beta cells against apoptosis. The authors also mentioned in the discussion that the treatment with IL-1Ra can reduce insulin dependence in the T1D children. Thus, the insulin levels in these KO DM mice may be different from their WT controls. Again, insulin signaling influences the electrophysiological response of cardiac myocytes. How to distinguish the effects of global attenuation in inflammation in the KO mice from the local effects of anti-arrhythmic in the myocardium?

Answer: Regarding the question of insulin levels, we would like to refer the reviewer to point (2) above, where we have answered this in detail.

Reviewer 1 is right about the possibility that the attenuation of inflammation found in some KOs may protect pancreatic beta cells against apoptosis, thus preventing the decrease in insulin level. Since the lack of insulin signaling can reduce I_{to} and produce QTc and AP prolongation, as shown previously in cardiac-specific insulin receptor KO mice⁶, we have measured the circulating insulin levels. Most diabetic KOs have decreased insulin levels, though the difference between diabetic and non-diabetic KOs does not always reach statistical significance. However, diabetic *Nlrp3*^{-/-} mice present insulin levels similar to non-diabetic *Nlrp3*^{-/-}. In these mice we cannot rule out the participation of insulin signaling in the prevention of I_{to} reduction and consequently QTc and AP prolongation.

We also have integrated this important point also into the revised discussion section (page 8; line: 306).

The absence of insulin signaling can reduce I_{to} current, resulting in AP and QT prolongation⁶. Nevertheless, the circulating insulin levels in diabetic *Il-1r*^{-/-} were significantly decreased compared to non-diabetic counterparts, while a strong trend towards decreased levels was found in diabetic *Tlr2*^{-/-} and *Casp1*^{-/-} mice,

even though it was not statically different. Only diabetic *Nlrp3*^{-/-} mice presented circulation insulin levels similar to non-diabetic controls. Although we can not exclude the possibility that preserved circulating insulin levels account for the prevention of electrophysiological abnormalities in diabetic *Nlrp3*^{-/-} mice, this is certainly not the case in *Il-1r*^{-/-}, *Tlr2*^{-/-} and *Casp1*^{-/-} mice. These results reflect the limitation of our study where the systemic and local anti-inflammatory effects in the KO animals are not discernable using our global KO mice models. These results reflect the limitation of our study where the systemic and local anti-inflammatory effects in the KO animals are not discernable using our global KO mice models.

5. It has been reported that *IL-1b* reduces L type Ca^{2+} channel current density in cardiomyocytes. (*J Biol Chem.* 2014, 289(32):21896-908.) How is the *ICaL* in the models tested in this study?

Answer: The L type Ca^{2+} channel current density was not measured in this study, since previous studies demonstrated that this current is depressed in diabetes⁷⁻⁹. This decrease would result in shortening of action potential duration (APD) and therefore most likely not underlie enhanced ventricular electrical vulnerability. In our study, we have focused on cellular targets underlying the prolonged action potential duration and hence increased VT incidence.

Minor points:

1. What is the expression level of *Tlr2* in the wild type mice?
Does it increase in the DM WT mice?

Answer: In order to answer this question new experiments have been performed. The results obtained here did not show any difference in TLR2 expression, at least concerning cardiac resident macrophages, when diabetic mice were compared to non-diabetic mice. These new data are shown in new Supplementary **Fig. 6**.

Additionally the following sentence was added to the revised results section:

TLR2 expression in cardiac macrophages was similar between wild type, non-diabetic and diabetic mice (Supplementary Fig. 6).

2. How about the expression level of other cytokines, such as TNF α ?

Answer: Systemic TNF- α values were higher in diabetic WT mice, than in non-diabetic WT mice ($P > 0.05$).

Reviewer #2 (expert in inflammasome and diabetic cardiomyopathy)

Remarks to the Author:

Monnerat et al present a manuscript exploring the role of IL-1b in diabetes induced cardiac arrhythmias in mice. They show that macrophages expressing TLR2 sense endogenous ligands triggering the expression of the NLRP3 inflammasome and IL-1b. Through unknown signals the NLRP3 inflammasome is activated causing release of IL-1b. It appears that IL-1b in its own right promotes the arrhythmia. Using genetically deficient mice they prove the importance of this pathway. Finally they show the translatable ability of their findings as administration of Anakinra or MCC950 could reduce diabetes induced arrhythmias.

Comments:

1. Figure 1 - the authors went directly to look at TLR2 but it is unclear why TLR2 and not TLR4 or RAGE were not also considered? While the TR2 KO mice are clearly protected, there show to be a better rationale why this was examined only.

Answer: We thank the reviewer for raising this point. We have focused on TLR2, because earlier studies have suggested a key role of TLR2 for several cardiac diseases, including cardiac arrhythmias^{10,11}. In addition, it was also demonstrated that lack of TLR2 prevents the increment of IL-1 β levels in different animal models¹². Interestingly, even though IL-1 β is a key cytokine broadly described in

several diseases that impair cardiac function, the potential role of IL-1 β in altering cardiac properties, at least in our view, has not been intensely investigated so far. We have explained the rationale for focusing on TLR2 in more detail in the revised version of text (page: 3; line: 85). We agree with the reviewer that TLR4 and/or RAGE could also play a role and it would be therefore interesting to explore this aspect in future studies.

Also is TLR2 unregulated on cardiac macrophages from diabetic animals?

Answer: We thank the reviewer for bringing this up and we have performed new experiments to address this question. Our results reveal that there is no difference in cardiac macrophage TLR2 expression in diabetic vs. non-diabetic WT mice. These new data are incorporated into Supplementary **Fig. 6**.

The following sentence has been added to the revised results section of the ms (page: 6; line: 212):

TLR2 expression in cardiac macrophages was similar in non-diabetic and diabetic WT mice (New Supplementary Fig. 6).

2. Figure 1i: The western blot is difficult to assess can the quality be improved? Can you also perform gene expression to compliment this result?

Answer: Total levels of IL-1 β are usually assessed by ELISA, but western blot is for now the only tool to assess mature IL-1 β . The western blot for mature IL-1 β is relatively straight forward, when extracts of isolated cells such as monocytes/macrophages or fibroblasts are processed. However, it turns out to be more challenging, when tissue extracts are used: Although good liver western blots can be easily achieved, we found that for heart samples this is much more difficult and our many attempts to further improve the quality of these western blots failed; interestingly we could also not find in the literature other work reporting IL-1 β protein levels in heart, indicating that it might indeed be

challenging to determine such protein levels in this organ. In any case, we have repeated the blot with similar results 4 times and would therefore consider it of enough good quality/reliability to quantify mature IL-1 β in the heart.

In addition, as suggested by the reviewer, total cardiac IL-1 β mRNA was assessed by qPCR assay and no differences in IL-1 β mRNA expression could be observed comparing WT and *Tlr2*^{-/-} non-diabetic vs their diabetic counterparts, respectively. This new information has been included in the revised version of the manuscript (page: 4; line: 109).

Supplementary Figure 6; Emiliano Medei

3. In all of these genetic and pharmacological interventions was there any changes in heart weight or hypertrophy?

Answer: We thank the reviewer for bringing up this important point. Both heart weight/body weight or heart weight/tibia length ratios, which are extensively used as cardiac hypertrophic index, were very similar among all experimental groups studied. For wild type and *Tlr2*^{-/-} mice these data are reported in **Supplementary Fig. 2**, for *Nlrp3*^{-/-} and *Casp1*^{-/-} in **Supplementary Table 2** and for *IL-1r*^{-/-} mice in **Table 3**.

The following new Supplementary data have been added:

Supplementary Table 1. Blood Glucose and Insulin levels

WT	WT+DM	P value	TLR2 ^{-/-}	TLR2 ^{-/-} +DM	P value

Glucose (mg/dL)	102.6 ± 5.2	343.3 ± 37.7	0.0006	95.29 ± 2.9	471.0 ± 26.9	< 0.0001
Insulin (μIU/mL)	12.7 ± 0.7	10.1 ± 0.9	0.0471	11.0 ± 0.5	9.5 ± 0.4	0.0741

n: WT: 7; WT+DM: 8; *Tlr2*^{-/-}: 6; *Tlr2*^{-/-}+DM: 6.

Supplementary Table 2. Blood Glucose and Insulin levels and Cardiac Biometry

	Nlrp3 ^{-/-}	Nlrp3 ^{-/-} +DM	P value	Casp1 ^{-/-}	Casp1 ^{-/-} +DM	P value
Glucose (mg/dL)	132.6 ± 9.4	361.0 ± 20.3	< 0.0001	122.8 ± 6.0	337.4 ± 0.7	0.0006
Insulin (μIU/mL)	12.7 ± 1.4	10.2 ± 0.8	0.1761	16.7 ± 2.5	10.9 ± 1.0	0.0542
HW/BW (mg/g)	6.4 ± 0.4	5.9 ± 0.2	0.3894	6.5 ± 0.6	6.4 ± 0.3	0.7787
HW/TL (mg/cm)	8.2 ± 0.6	7.2 ± 0.3	0.1449	9.1 ± 0.7	8.4 ± 0.6	0.4538

For glucose and Insulin: n: *Nlrp3*^{-/-}: 7; *Nlrp3*^{-/-}+DM: 6; *Casp1*^{-/-}: 7; *Casp1*^{-/-}+DM: 7.
 For Cardiac Biometry: n: *Nlrp3*^{-/-}: 9; *Nlrp3*^{-/-}+DM: 11; *Casp1*^{-/-}: 9; *Casp1*^{-/-}+DM: 13.

Supplementary Table 3. Blood Glucose and Insulin levels and Cardiac Biometry

	IL-1r^{-/-}	IL-1r^{-/-}+DM	P value
Glucose (mg/dL)	143.3 ± 13.1	387.7 ± 25.3	< 0.0001
Insulin (μIU/mL)	12.9 ± 1.3	9.0 ± 0.9	0.0403
HW/BW (mg/g)	5.4 ± 0.2	5.2 ± 0.3	0.4400
HW/TL (mg/cm)	6.5 ± 0.2	7.1 ± 0.4	0.2872

For glucose and Insulin: **n:** *IL-1r^{-/-}*: 6; *IL-1r^{-/-}+DM*: 6. For Cardiac Biometry: **n:** *IL-1r^{-/-}*: 5; *IL-1r^{-/-}+DM*: 5.

The following text has been added to the revised results section (page: 6, line: 233 and page: 7, line: 258):

- i- The heart weight/body weight as well as the heart weight/tibia length ratios were similar among study groups (Supplementary Table 2).
- ii- No difference in bodyweight/cardiac weight as well as the heart weight/tibia length ratios was observed between *IL-1r^{-/-}* diabetic and non-diabetic mice (Supplementary Table 3).

4. Figure 5J and Figure 6I: There appears to be a replication of the data in these two figures - the control groups are the same for both. If these interventions share the same control group they should be graphed in the same figure.

Answer: We would like to thank the reviewer for making this point. In fact these interventions share the same control group. In order to maintain the sequence of the results, the control data are only shown in Fig. 6j, whereas we have taken these out from Fig. 7i and explained this in the results section and the respective legend (Fig. 7f-j) of the revised manuscript (page: 23; line: 501).

Fig. 7f-j – This experiment shares the same control group as Fig. 6j - and Supplementary Fig. 8c,d.

Reviewer #3 (expert in cardiac arrhythmia)

Remarks to the Author:

This is an interesting and potentially important paper that provides convincing evidence for an association between hyperglycemia, activation of IL-1 β sourced from macrophages, and arrhythmias. The strengths of this study are the insights into the underexplored connections between non-myocyte inflammatory cells and signaling and myocardial disease and the clinical/translational implications of these findings. The weaknesses are the failure to 'connect the dots' between various phenomena: is loss of I_{to} a critical step in the arrhythmias? If so it is likely less relevant to larger mammals (including humans) because I_{to} is far more important in rodents where I_{to} is critical for their abbreviated AP duration. Is AP prolongation necessary to activate CaMKII or does increased oxidation and/or O-GlcNAcylation activate CaMKII? Although CaMKII appears essential to the proposed arrhythmogenic pathway, its role is only explored in cursory fashion (e.g. in rodent and not iPS cells and only using KN-93). KN-93 has numerous off target actions, including ionic currents with potential relevance to the observed arrhythmia phenotypes. In contrast, these off target actions are not shared by the 'control' compound KN-92. Thus, cleaner approaches to CaMKII inhibition are required. Fortunately, many options are available including mice lacking CaMKII δ , mice with myocardial transgenic expression of AC3-I (I believe these are in the laboratory of the nearby La Plata group), mice with oxidation-resistant CaMKII, shRNA knock down for cellular studies, etc.

Answer: We would like to thank the reviewer for his very positive remarks on our work and his constructive criticisms. We agree with the reviewer concerning the important role of CaMKII and have therefore performed new experiments to strengthen this part of the story (below).

Although we cannot unequivocally prove that decrease in I_{to} is critical for the generation of the arrhythmias, this decrease could contribute, at least in part, to the prolongation of AP duration. This has been observed in DM WT mice and also been proven to be a relevant arrhythmogenic substrate in large mammals¹³. Thus, we think that prolongation of the AP duration in combination with the observed enhanced SR Ca²⁺ release are pro-arrhythmogenic substrates through which IL-1 β could promote ventricular arrhythmias in DM.

We agree with the reviewer that in the setting of DM, CaMKII could be activated by other mechanisms in addition to oxidation, such as Ca²⁺-dependent activation due to enhanced Ca²⁺ entry resulting from AP prolongation or by GLcNAcylation due to elevated glucose levels. We show that direct incubation of cardiac myocytes with IL-1 β activates CaMKII (increased p-CaMKII and increased OxiCaMKII) and prolongs AP. However, IL-1 β did not increase the intracellular Ca²⁺ transient in cardiac myocytes. Furthermore, as IL-1 β is a downstream signaling molecule in DM it is not expected to increase glucose levels. Taking

into account this contention and the fact that we did not observe a significant increase intracellular Ca^{2+} levels in the presence of $\text{IL-1}\beta$, we conclude that the main mechanism underlying $\text{IL-1}\beta$ -induced CaMKII activation would be an oxidation-dependent activation.

We have, as suggested by the reviewer, performed new experiments using KN92, which lacks the off-target effects of KN93: WT cardiomyocytes were exposed to $\text{IL-1}\beta$ together with KN92 (2.5 $\mu\text{mol/L}$) for 24 h. In new Fig. 3d the results of these experiments are shown, namely that cardiomyocyte exposure to a combination of $\text{IL-1}\beta$ -KN92 showed a greater NSE, when compared to the exposure of $\text{IL-1}\beta$ and KN93. These data reinforce the idea that $\text{IL-1}\beta$ -induces greater NSE activating CaMKII.

In order to clearly demonstrate the involvement of CaMKII we have followed the very helpful advise of the reviewer and used transgenic mice with either cardiomyocyte specific expression of AC3-I or autocamtide 3 control peptide (AC3-C). Dispersed cardiomyocytes from AC3-I mice were incubated for 24 h in the presence or in the absence (control condition) of $\text{IL-1}\beta$. As shown in new Fig 3.c,d, these experiments underscored our working hypothesis, namely that CaMKII activation leads to increase in $\text{IL-1}\beta$ -induced number of spontaneous contractions when applying our stimulation pattern. In line with this, isolated cardiomyocytes from AC3-C mice showed higher number of NSE after 24 h of $\text{IL-1}\beta$ incubation compared to cardiomyocytes obtained from AC3-C cells in the absence of $\text{IL-1}\beta$ exposure.

In addition, as suggested by the reviewer, we have also performed new experiments to better explore the translational impact of our work. Therefore, we have explored in hiPS-CM, whether IL-1 β activation of CaMKII could underlie the observed IL- β -induced electrophysiological changes. For this purpose hiPS-CM were incubated for 24 h with IL-1 β (10 ng/mL) in the presence or in the absence of the selective CaMKII inhibitor autocamtide-2-related inhibitory peptide (AIP). New Fig 3e,f in the main manuscript, shows that the CaMKII inhibition was able to prevent the IL-1 β -induced cardiac field potential prolongation in hiPS-CM. These data are also in full agreement with our rodent data demonstrating that activation of CaMKII plays a key role in the IL-1 β -induced higher NSE.

References

1. Sordillo, P.P., Sordillo, D.C. & Helson, L. Review: The Prolonged QT Interval: Role of Pro-inflammatory Cytokines, Reactive Oxygen Species and the Ceramide and Sphingosine-1 Phosphate Pathways. *In Vivo* **29**, 619-636 (2015).
2. Li, Y.H. & Rozanski, G.J. Effects of human recombinant interleukin-1 on electrical properties of guinea pig ventricular cells. *Cardiovasc Res* **27**, 525-530 (1993).
3. Mitrokhin, V.M., Mladenov, M.I. & Kamkin, A.G. IL-1 provokes electrical abnormalities in rat atrial myocardium. *Int Immunopharmacol* **28**, 780-784 (2015).
4. Epelman, S., *et al.* Embryonic and adult-derived resident cardiac macrophages are maintained through distinct mechanisms at steady state and during inflammation. *Immunity* **40**, 91-104 (2014).
5. Dasu, M.R., *et al.* TLR2 expression and signaling-dependent inflammation impair wound healing in diabetic mice. *Lab Invest* **90**, 1628-1636 (2010).
6. Lopez-Izquierdo, A., *et al.* The absence of insulin signaling in the heart induces changes in potassium channel expression and ventricular repolarization. *Am J Physiol Heart Circ Physiol* **306**, H747-754 (2014).
7. Chattou, S., Diacono, J. & Feuvray, D. Decrease in sodium-calcium exchange and calcium currents in diabetic rat ventricular myocytes. *Acta Physiol Scand* **166**, 137-144 (1999).
8. Wang, D.W., Kiyosue, T., Shigematsu, S. & Arita, M. Abnormalities of K⁺ and Ca²⁺ currents in ventricular myocytes from rats with chronic diabetes. *Am J Physiol* **269**, H1288-1296 (1995).
9. Lu, Z., *et al.* Decreased L-type Ca²⁺ current in cardiac myocytes of type 1 diabetic Akita mice due to reduced phosphatidylinositol 3-kinase signaling. *Diabetes* **56**, 2780-2789 (2007).
10. Mersmann, J., *et al.* Toll-like receptor 2 signaling triggers fatal arrhythmias upon myocardial ischemia-reperfusion. *Crit Care Med* **38**, 1927-1932 (2010).
11. Ichiki, H., *et al.* The role of infection in the development of non-valvular atrial fibrillation: up-regulation of Toll-like receptor 2 expression levels on monocytes. *J Cardiol* **53**, 127-135 (2009).
12. Wang, L., *et al.* Inhibition of Toll-like receptor 2 reduces cardiac fibrosis by attenuating macrophage-mediated inflammation. *Cardiovasc Res* **101**, 383-392 (2014).
13. Volders, P.G., *et al.* Cellular basis of biventricular hypertrophy and arrhythmogenesis in dogs with chronic complete atrioventricular block and acquired torsade de pointes. *Circulation* **98**, 1136-1147 (1998).

REVIEWERS' COMMENTS:

Reviewer #1 (Remarks to the Author):

The authors have addressed most of my concerns. One remaining point is that, as the mRNA level of TGF-1b is not changed, and even showed a trend of decrease in the DM hearts; the induction of TGF-1b by Tlr2 is supported just by western blot. However, the quality of the blot is not satisfactory and n is only 3. The authors argued that the quality of the blot cannot be further improved. Can the sample size be increased to ensure the reliability of this result?

Reviewer #2 (Remarks to the Author):

No further comments

Reviewer #3 (Remarks to the Author):

The authors have been highly responsive and provided new and convincing data that satisfactorily address my reservations. The role of oxidized CaMKII has been established in the STZ model of DM (Luo JCI 2013), and this publication should be cited given the centrality of this topic to the author's research.

Poin to point answer to Reviewers

REVIEWERS' COMMENTS:

Reviewer #1 (Remarks to the Author):

The authors have addressed most of my concerns. One remaining point is that, as the mRNA level of TGF-1b is not changed, and even showed a trend of decrease in the DM hearts; the induction of TGF-1b by Tlr2 is supported just by western blot. However, the quality of the blot is not satisfactory and n is only 3. The authors argued that the quality of the blot cannot be further improved. Can the sample size be increased to ensure the reliability of this result?

Answer: in order to answer the Reviewer question the authors increased the sample size, from N = 3 to N = 6. Thus a new Western Blot experiment was conducted with samples from other 3 different hearts from each experimental group (WT, WT-diabetic, *Tlr2*^{-/-}, *Tlr2*^{-/-} diabetic). As observed in the following new Figure the experiments conducted with the new samples confirm the previous results:

Additionally, the correspondent uncropped Western Blot appears as Supplementary Figure 9.

Reviewer #3 (Remarks to the Author):

The authors have been highly responsive and provided new and convincing data that satisfactorily address my reservations. The role of oxidized CaMKII has been established in the STZ model of DM (Luo JCI 2013), and this publication should be cited given the centrality of this topic to the author's research.

Answer: the authors agree with the reviewer, thus the suggested Reference was added into the main manuscript under number 30.